# Large transverse thermoelectric effect induced by the mixed-dimensionality of Fermi surfaces

Hikari Manako [1], Shoya Ohsumi [1], Yoshiki J. Sato [1,3] ✉, R. Okazaki [1] & D. Aoki[2]

Transverse thermoelectric effect, the conversion of longitudinal heat current into transverse electric current, or vice versa, offers a promising energy harvesting technology. Materials with axis-dependent conduction polarity, known as $p \times n$-type conductors or goniopolar materials, are potential candidate, because the non-zero transverse elements of thermopower tensor appear under rotational operation, though the availability is highly limited. Here, we report that a ternary metal $LaPt_2B$ with unique crystal structure exhibits axis-dependent thermopower polarity, which is driven by mixed-dimensional Fermi surfaces consisting of quasi-one-dimensional hole sheet with out-of-plane velocity and quasi-two-dimensional electron sheets with in-plane velocity. The ideal mixed-dimensional conductor $LaPt_2B$ exhibits an extremely large transverse Peltier conductivity up to $|\alpha_{yx}| = 130$ A K$^{-1}$ m$^{-1}$, and its transverse thermoelectric performance surpasses those of topological magnets utilizing the anomalous Nernst effect. These results thus manifest the mixed-dimensionality as a key property for efficient transverse thermoelectric conversion.

Thermoelectricity, a mutual conversion between heat and charge currents, has an essential role in sustainable energy technology to utilize the waste heat ineluctably exhausted from the industrial society[1–3]. In addition to the longitudinal conversion known as the Seebeck and the Peltier effects (Fig. 1a), recent progress on the transverse thermoelectric effect[4–7] has opened a new avenue for efficient thermoelectric conversion (Fig. 1b, c) because the transverse devices are able to be designed to cover the entire area of a heat source with highly-flexible compatibility[8–10]. In the transverse thermoelectricity, the off-diagonal term of the thermopower tensor $\hat{S}$ in the relation $\mathbf{E} = \hat{S}\nabla T$, where $\mathbf{E}$ and $\nabla T$ respectively express the electric field and the temperature gradient, is essential. The ordinary Nernst effect is a prime example of inducing the off-diagonal elements and is largely enhanced for clean degenerate semiconductors[11–14]. Moreover, the anomalous Nernst effect (ANE) has been intensively studied in nontrivial topological magnets in which the emergent field of the Berry

curvature plays a critical role in triggering the large transverse signals[8,15–19], while the external magnetic field or magnetization is indispensably required in these Nernst-based transverse thermoelectric effects (Fig. 1b).

The $p \times n$-type material[5], also known as the goniopolar material[7,20], in which opposite carrier polarity appears depending on the crystallographic axis, offers the simplest solution to achieve efficient transverse thermoelectric conversion that operates at zero magnetic field[10,21]. The schematic figure of a simple goniopolar conductor exhibiting the axis-dependent conduction polarity between the crystallographic $i$ and $j$ axes ($S_{ii} = -S_{jj} = S > 0$) is illustrated in Fig. 1c. When the temperature gradient $\nabla_x T$ is applied to $x$-axis direction, which is the direction rotated by an angle $\phi$ from crystallographic $i$-axis, the transverse electric field $E_y = S_{yx}\nabla_x T = [(S_{ii} - S_{jj})\cos\phi\sin\phi]\nabla_x T$ can be obtained based on the rotation of the thermopower tensor. The goniopolar conductors show thermopower with opposite polarities,

[1]Department of Physics and Astronomy, Tokyo University of Science, Noda, Japan. [2]Institute for Materials Research, Tohoku University, Oarai, Ibaraki, Japan. [3]Present address: Graduate School of Science and Engineering, Saitama University, Saitama, Japan. ✉e-mail: yoshikisato@mail.saitama-u.ac.jp

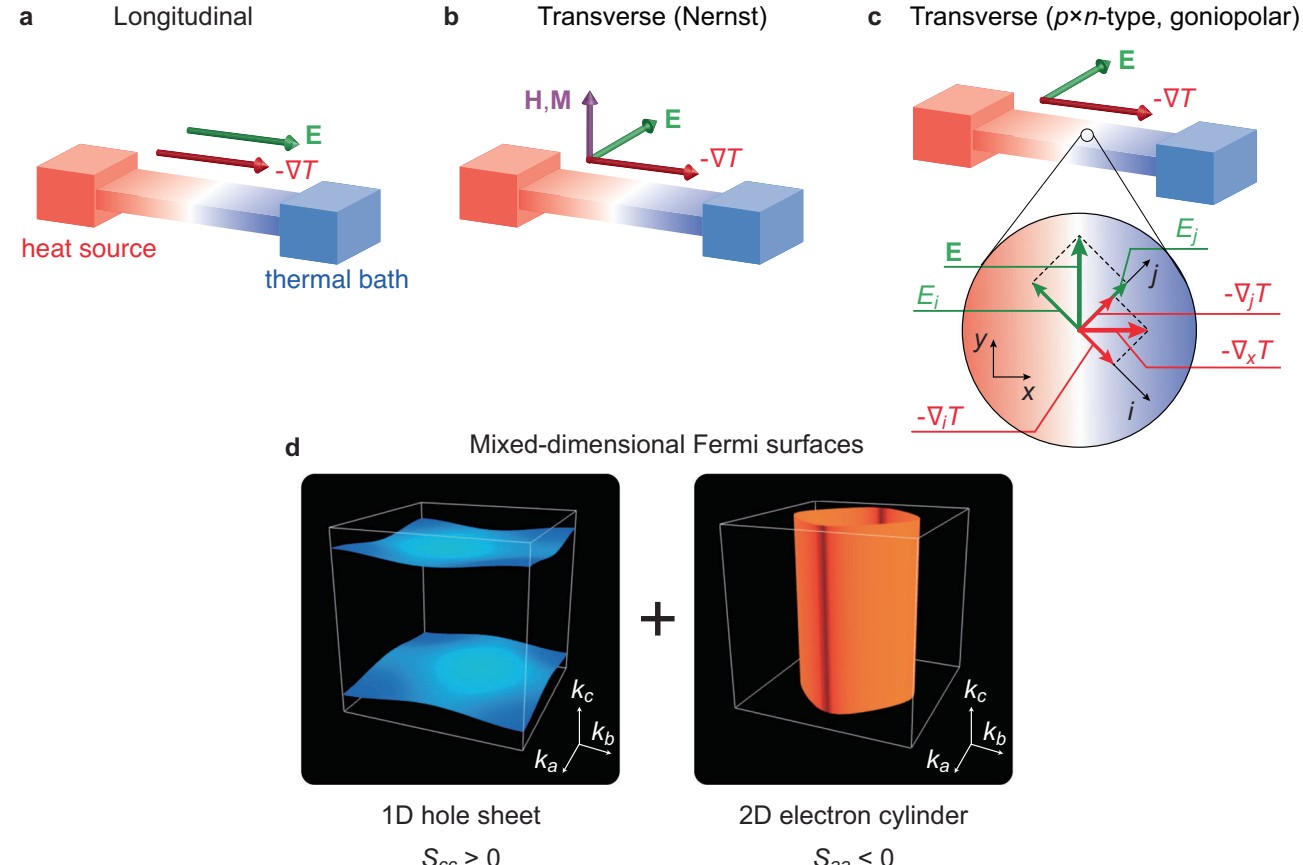

**Fig. 1 | Concept of mixed-dimensionality for transverse thermoelectrics.** In the longitudinal (**a**) and transverse (**b**, **c**) thermoelectric effects, the electric field **E** appears parallel and perpendicular to the temperature gradient $-\nabla T$, respectively. **b** Ordinary and anomalous Nernst effects induced by the external magnetic field **H** or magnetization **M**. **c** Transverse thermoelectric effect in $p \times n$-type (goniopolar) conductor. Temperature gradient is applied to a direction rotated from the crystallographic axis. Owing to the opposite polarities for each axis, the electric field is induced along the transverse direction. **d** Schematic illustration of the mixed-dimensional Fermi surfaces. Two large Fermi surfaces, a one-dimensional (1D) hole sheet with a normal vector along the c-axis and a two-dimensional (2D) electron cylinder with a normal vector along the ab-planes direction, are considered. As a result of these combinations, though hybridization should occur at the crossing points, a goniopolar metal with positive c-axis thermopower ($S_{cc} > 0$) and negative in-plane thermopower ($S_{aa} < 0$) may be obtained.

and thus the transverse thermopower $S_{yx}$ is largely enhanced. However, design principles for such goniopolar materials have not yet been established. For various recently found goniopolar conductors[20–24], as well as several elements[25] and organics[26], the origin is simply referred to as a multi-band effect with anisotropic carrier mobilities. A single-band goniopolar picture[20] with opposite directional band curvatures is convincingly appropriate for several layered compounds[20,27,28]. The anisotropic band structures are important in the emergence of goniopolar conduction[5,20,22–24,29], whereas the resulting transport anisotropy in such layered systems imposes a severe constraint on the optimal heat-current angle. Moreover, small anisotropic Fermi pockets in semi-metallic compounds or semiconductors often cause a complicated temperature dependence of thermopower, restricting the applicable temperature range for transverse thermoelectric conversion. In these respects, an ideal goniopolar material should have isotropic transport coefficients while exhibiting thermopower with opposite polarities for different crystallographic axes in a wide temperature range.

Here, we establish the concept of mixed-dimensionality of Fermi surfaces in metals as the design principles of efficient transverse thermoelectric materials. A minimal model for the mixed-dimensional conductor with the axis-dependent conduction polarity is schematically depicted in Fig. 1d. Instead of small anisotropic Fermi pockets as seen in degenerate semiconductors, we consider two large Fermi surfaces with opposite polarities as a key property: a one-dimensional

(1D) hole sheet with normal vector along the c-axis and a two-dimensional (2D) electron cylinder with normal vector along the in-plane direction. As a result of these combinations, although a hybridization should occur at the crossing points to modify the sheet shape, we expect the axis-dependent conduction polarity in which the positive and negative thermopower, respectively, appear along the out-of-plane and in-plane directions. Moreover, simple $T$-linear dependence of the thermopower is expected in metallic compounds for both crystallographic directions, in sharp contrast to the case of the degenerate semiconductors with a small Fermi pocket to show the complex temperature dependence of thermopower and pronounced sample dependence. Tungsten disilicide $WSi_2$ is one of the candidate materials with mixed-dimensional bulk Fermi surfaces[30], although its transverse thermoelectric properties have not been investigated yet. In this study, we report the discovery of an ideal material with axis-dependent thermopower polarity based on the mixed-dimensionality of the Fermi surface and demonstrate the high transverse thermoelectric performance of the mixed-dimensional material through direct measurements of transverse thermopower.

In order to materialize such a mixed-dimensional conductor based on the concept of mixed-dimensionality, we here study a ternary compound $LaPt_2B$, a non-magnetic metal in a series of the rare-earth-based chiral magnets $RPt_2B$ ($R$: rare-earth elements, space group $P6_222$ or $P6_422$)[31–33]. This material possesses a screw axis along the c-axis (Fig. 2a, b) to form a monoaxial spin interaction in the chiral magnets

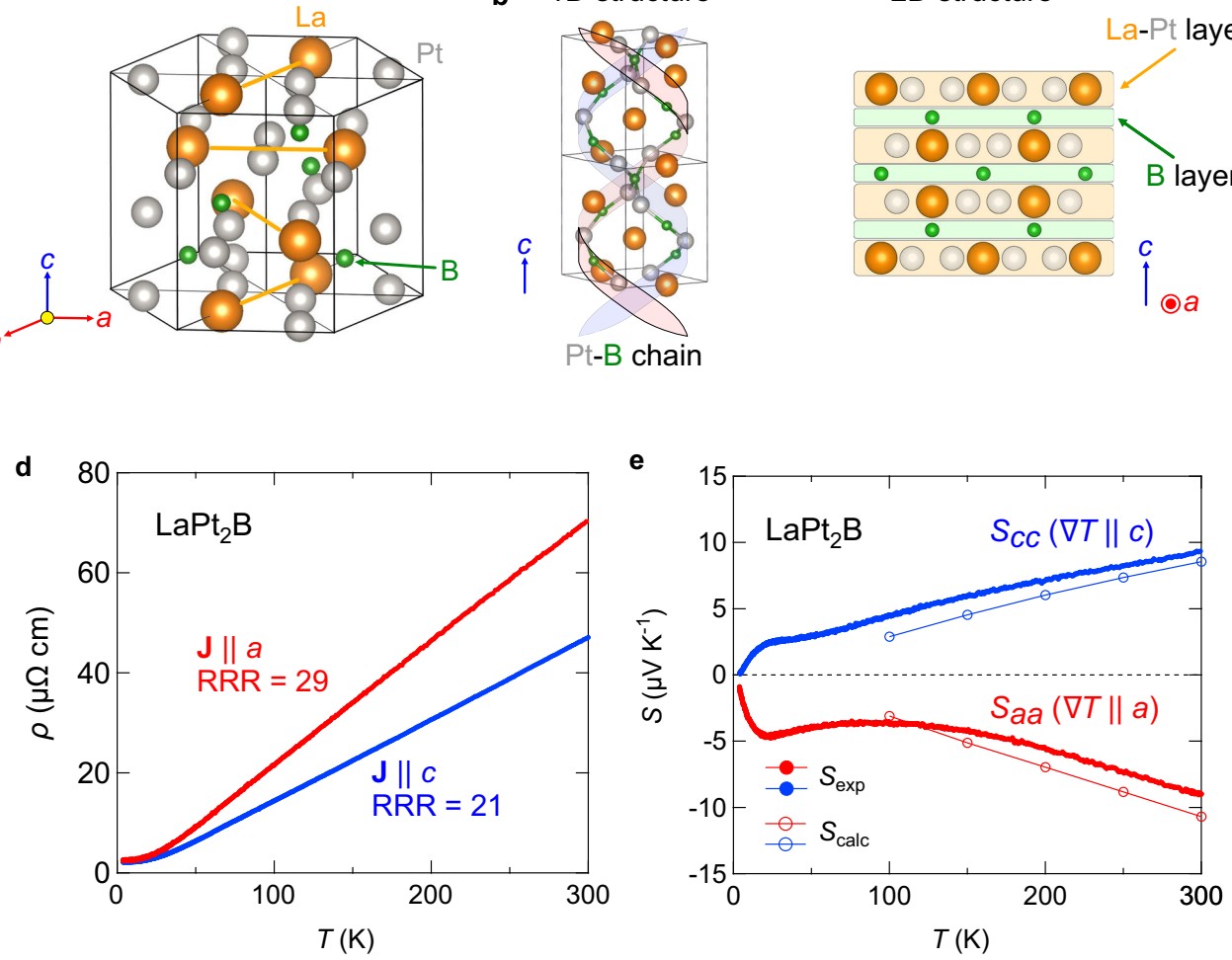

**Fig. 2 | Transport properties and axis-dependent thermopower polarity in chiral crystal LaPt₂B. a** Crystal structure of LaPt₂B. **b** Chain-like Pt-B connection along the screw-axis. **c** Side view of the crystal structure. The La-Pt and B layers alternately stack along the *c*-axis direction. **d** Temperature dependence of the electrical resistivity along the *a*-axis (red) and *c*-axis (blue) directions. **e** Temperature dependence of the thermopower along the *a*-axis (red) and *c*-axis (blue) directions. The experimental data and the calculation results are displayed with solid and open symbols, respectively.

*R*Pt₂B[34,35], indicating an orbital hybridization along the *c*-axis direction. Furthermore, it shows a layered structure consisting of La-Pt and B layers stacking along the *c*-axis (Fig. 2c), in which the extended orbitals within the layers are expected. Thus, one-dimensional axial and two-dimensional layered structural properties coexist in this material.

## Results

### Transport properties and axis-dependent thermopower polarity

We examine the anisotropic transport properties in LaPt₂B single crystals. The electrical resistivity shows a metallic temperature variation characterized by conventional electron-phonon scattering (Fig. 2d). The small low-temperature resistivity and the large residual resistivity ratio (RRR) for both *a*-axis and *c*-axis directions indicate a high purity of the present single crystals. The temperature dependence of the thermopower of LaPt₂B is shown in Fig. 2e. It is noteworthy that a goniopolar conduction, positive *c*-axis and negative *a*-axis thermopower, is clearly observed in a wide temperature range below room temperature. The high quality of the present crystals is also evidenced by a small phonon-drag peak structure at low temperatures[36]. These transport properties demonstrate that the present material may be a potential candidate for efficient transverse thermoelectricity using the axis-dependent conduction polarity. LaPt₂B exhibits the axis-dependent thermopower polarity, while the

Hall coefficient shows the same sign for both crystallographic axes (Fig. S11). To avoid confusion caused by the use of terms such as *p* × *n*-type or goniopolar conductors, which imply the axis-dependent conduction polarity in both the Hall coefficient and thermopower, here we define materials exhibiting axis-dependent thermopower polarity based on the mixed-dimensionality of Fermi surfaces as mixed-dimensional conductors.

### Mixed-dimensional Fermi surfaces

Based on first-principles calculations, a microscopic origin for such axis-dependent thermopower polarity in LaPt₂B is then discussed. The thermopower calculated using the Boltzmann equations within a relaxation time approximation qualitatively agrees with the experimental data (Fig. 2e), indicating that the observed axis-dependent thermopower polarity is basically understood within a band picture. We thus consider the Fermi surface topology. Indeed, the calculated Fermi surfaces exhibit mixed-dimensional nature (Fig. 3a). We have obtained five Fermi surfaces labeled α, β, γ, δ, and ε within the scalar relativistic calculations. The hole-like α sheet has a large surface area with normal vectors along the *c*-axis, which significantly contributes to the positive thermopower along the *c*-axis direction. The electron-like β sheet with largely warped cylindrical shape is rather three-dimensional. On the other hand, the electron-like γ sheet forms a

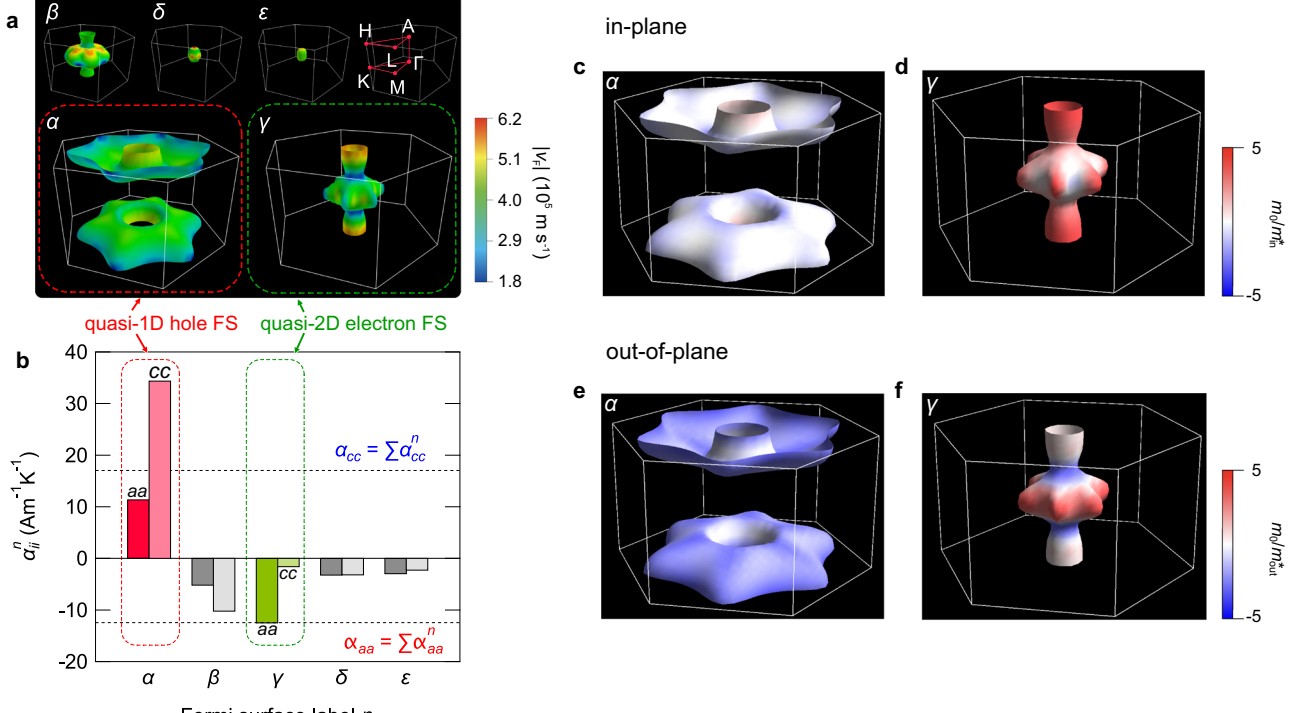

**Fig. 3 | Mixed-dimensional Fermi surfaces of LaPt$_2$B. a** Fermi surface calculated in the absence of spin-orbit coupling. Five sheets are labelled $\alpha$, $\beta$, $\gamma$, $\delta$, and $\varepsilon$. The color scale indicates the magnitude of the Fermi velocity $v_F$. High symmetry points in the hexagonal lattice are also shown. The number of Fermi surfaces is doubled in the full relativistic calculations owing to the lack of inversion symmetry, and the overall shapes of the Fermi surfaces are essentially the same as those for the scalar relativistic calculations (Fig. S3). **b** Band-resolved partial Peltier conductivity $\alpha_{aa}^n$ (left bar) and $\alpha_{cc}^n$ (right bar) calculated at $T = 300$ K with a constant relaxation time $\tau = 10^{-14}$ s. The dashed lines represent the total Peltier conductivity for each axis direction. **c–f** Mixed-dimensional Fermi surfaces $\alpha$ and $\gamma$ showing the diagonal elements of the inverse mass tensor for in-plane direction (**c, d**) and out-of-plane direction (**e, f**).

quasi-2D cylindrical shape responsible for the negative thermopower along the *ab*-plane direction. The coexistence of hole and electron Fermi surfaces with different dimensionality is crucially important to realize the axis-dependent thermopower polarity in this system. The orbital-resolved Fermi surfaces analysis has further revealed that the in-plane La-Pt network and the screw-axis Pt-B connection are mainly responsible for the in-plane and *c*-axis conduction, respectively (Fig. S6).

To quantitatively analyze the mixed-dimensionality of Fermi surfaces of LaPt$_2$B, we evaluate band-resolved partial Peltier conductivity $\alpha_{ii}^n = \sigma_{ii}^n S_{ii}^n$ ($i = a, c$ and $n = \alpha, \beta, \gamma, \delta, \varepsilon$), where $\sigma_{ii}^n$ and $S_{ii}^n$ are the band-resolved partial electrical conductivity and thermopower, respectively (Fig. 3b). Since the total thermopower along the *i*-axis is given as $S_{ii} = \sum_n \alpha_{ii}^n / \sum_n \sigma_{ii}^n$, we now compare the partial Peltier conductivity $\alpha_{ii}^n$ for each band. For the *c*-axis conduction, as expected from the shape of the Fermi surface, the weight of the hole-like $\alpha$ sheet ($\alpha_{cc}^\alpha$) is significantly large and the anisotropy is evaluated as $\alpha_{cc}^\alpha / \alpha_{aa}^\alpha \approx 3.0 > 1$, indicating the one-dimensional character of the $\alpha$ sheet. In contrast, for the in-plane conduction, the electron-like cylindrical $\gamma$ sheet largely contributes to the total in-plane negative Peltier conductivity. Indeed, the $\gamma$ sheet has a small value of $\alpha_{cc}^\gamma / \alpha_{aa}^\gamma \approx 0.1 < 1$, reflecting its two-dimensional cylindrical shape.

Here we discuss the inverse mass tensor on the Fermi surfaces to examine in detail the relation between the anisotropy of the band-resolved partial Peltier conductivity and the Fermi surface topology. We evaluated in-plane component $1/m_{in}^* = (1/m_{aa}^* + 1/m_{bb}^*)/2$ and out-of-plane component $1/m_{out}^* = 1/m_{cc}^*$ of the inverse mass tensor. We plotted the strength of the inverse mass tensor along in-plane (Fig. 3c, d) and out-of-plane (Fig. 3e, f) directions on the mixed-dimensional Fermi surfaces $\alpha$ and $\gamma$. The quasi-1D hole Fermi surface $\alpha$ exhibits a significant contribution from the hole for the out-of-plane

direction compared to the in-plane direction. In contrast, the cylindrical part near the A point of the $\gamma$ Fermi surface exhibits a strong contribution from electrons for the in-plane direction, indicating the quasi-2D nature of the $\gamma$ Fermi surface. Thus, this computational quantitative analysis provides firm evidence of the mixed-dimensional nature of this material.

## Transverse thermopower

Having the established axis-dependent thermopower polarity of mixed-dimensional conductor, we demonstrate the transverse thermoelectric effect of LaPt$_2$B (Fig. 4a). We have measured simultaneously the voltage drop $\Delta V$ (Fig. 4b) and the temperature difference $\Delta T$ (Fig. 4c) for both longitudinal ($x$) and transverse ($y$) directions using four thermocouples (Fig. 4d). The temperature gradient $-\nabla T$ is homogeneously applied to 45° from the *a*-axis and *c*-axis. The transverse temperature difference $\Delta T_y$ is negligibly small as is also indicated from the homogeneous heat current flows seen in the thermographic image (Fig. 4e, f). As shown in Fig. 4a, clear transverse Seebeck voltage $\Delta V_y$ is observed owing to the axis-dependent thermopower polarity even in almost zero temperature difference along the transverse direction $\Delta T_y \approx 0$ K. The calculated $\Delta V_x$ and $\Delta V_y$ based on the rotation of the thermoelectric tensor using $S_{aa}$ and $S_{cc}$ (supplementary information) agree well with the experimental results obtained in this geometry (Fig. 4b). Assuming homogeneous temperature gradient and electric field, large transverse thermopower $S_{yx} = E_y / \nabla T_x = (\Delta V_y / w) / (\Delta T_x / L)$ is obtained (Fig. 4a). As expected from $S_{aa}(T)$ and $S_{cc}(T)$, *T*-linear transverse thermopower $S_{yx}$ with a small low-temperature phonon-drag peak structure is observed. The observed $S_{yx}$ in the mixed-dimensional LaPt$_2$B reaches as high as 10 μV K$^{-1}$ at around room temperature and exceeds the record room-temperature transverse thermopower in ANE-based system (6 μV K$^{-1}$ in Co$_2$MnGa[15]).

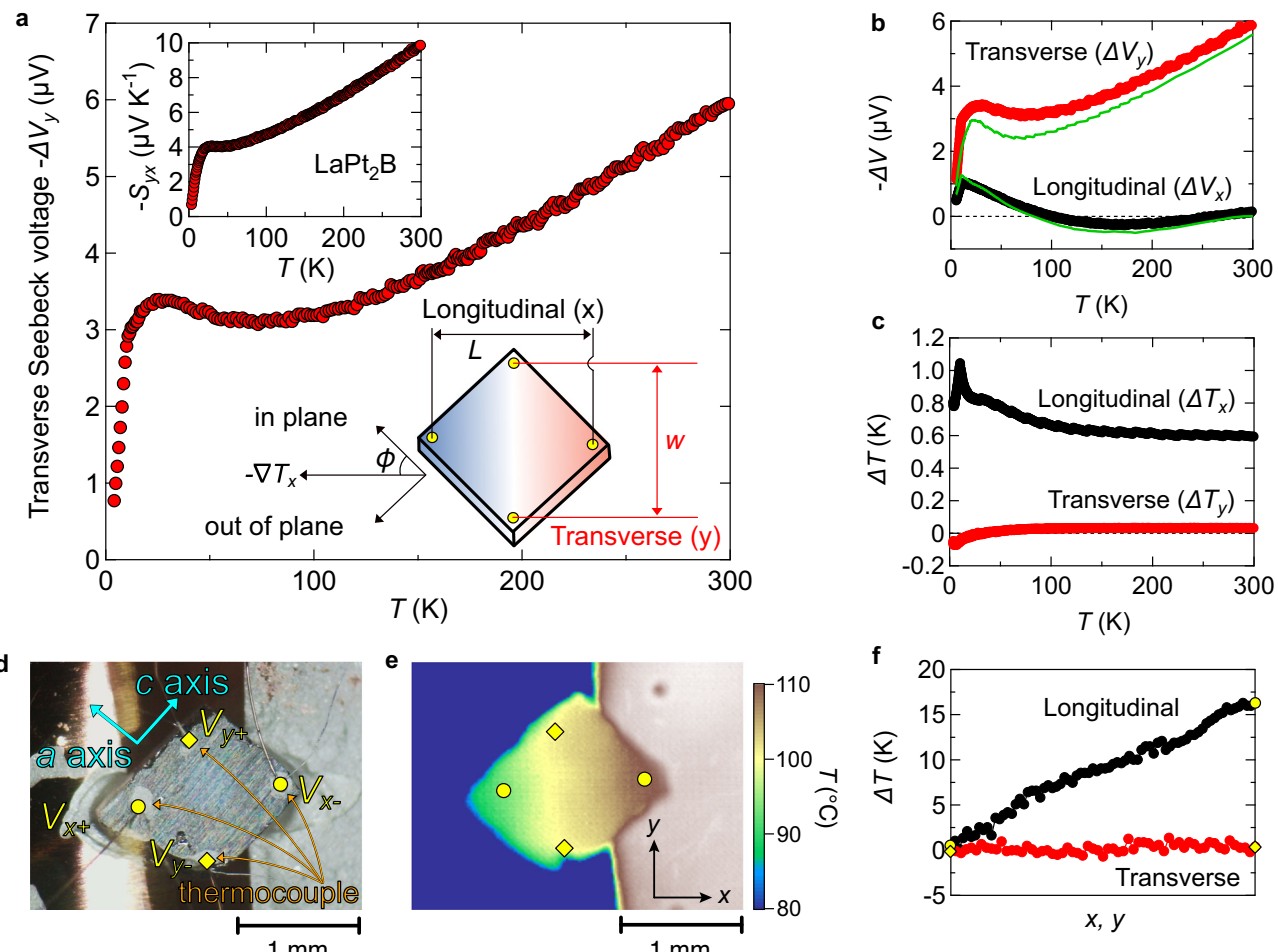

**Fig. 4 | Observation of transverse thermoelectric effect. a** Temperature dependence of the detected transverse Seebeck voltage $-\Delta V_y$ when a temperature gradient is applied at approximately 45° from the $a$-axis and $c$-axis. The schematic geometry of simultaneous measurement of longitudinal and transverse thermopower is shown in the main panel. The inset shows the estimated transverse thermopower $S_{yx}$ assuming homogeneous temperature gradient and electric field. **b**, **c** Temperature dependence of the voltage drop $\Delta V$ and temperature difference $\Delta T$ for both longitudinal ($x$) and transverse ($y$) directions. We have controlled the heater power so that the longitudinal temperature difference $\Delta T_x$ becomes approximately 0.6 to 1K. $\Delta V_x$ and $\Delta V_y$ calculated from the diagonal components $S_{aa}$ and $S_{cc}$ are shown by green solid lines, which agree well with the results obtained in this geometry. The green solid lines are calculated with the assumption that the heat flow angle is 45 degrees from the crystallographic axes. **d** One of the measured samples of LaPt$_2$B. As shown in the photograph, we have measured the temperature and voltage differences both for longitudinal and transverse directions using four thermocouples ($V_{x+/x-}$ and $V_{y+/y-}$). **e** The thermal image of the sample is shown in panel (**d**) at room temperature. The temperature gradient is uniform and applied almost perpendicular to the electrodes along the transverse direction. **f** The temperature variation between two thermocouples along the longitudinal and transverse directions.

The transverse thermoelectric effect of the mixed-dimensional conductor LaPt$_2$B is observed in zero magnetic field.

## Discussion

We compare the transverse Peltier conductivity $\alpha_{yx} = \sigma_{yx}S_{xx} + \sigma_{yy}S_{yx}$, a fundamental quantity to describe how large electric current is induced along the transverse $y$ direction by the applied temperature gradient along the $x$ direction, to those of other transverse thermoelectric materials including the ANE-based systems (Fig. 5a). In the present geometry where the heat current is applied at $\phi = 45°$ from the $a$-axis direction, $\alpha_{yx}$ is expressed using experimentally obtained $S_{xx}$ and $S_{yx}$ as

$$\alpha_{yx} = \frac{1}{2}(\sigma_{aa} + \sigma_{cc})S_{yx} + \frac{1}{2}(\sigma_{aa} - \sigma_{cc})S_{xx}. \qquad (1)$$

The transverse Peltier conductivity $\alpha_{yx}$ in the present material is fairly large compared with other transverse thermoelectric systems because of the large electrical conductivity and moderately large transverse thermopower. Astoundingly, the observed transverse

Peltier conductivity $|\alpha_{yx}|$ reaches as high as 130 A K$^{-1}$ m$^{-1}$ at $T = 15$ K, attributed to the significant contributions of both the large electrical conductivity and the phonon-drag enhancement of the Seebeck coefficient. LaPt$_2$B also exhibits a large transverse power factor in a wide temperature range below room temperature, as well as the giant Peltier conductivity (Fig. S13).

When the direction of the heat current is rotated from $a$-axis by an angle $\phi$, the optimal angle to maximize the efficiency, denoted as $\phi_{opt}$, can be expressed as follows[5]

$$\cos^2\phi_{opt} = \left(1 + \sqrt{\frac{\kappa_{cc}/\kappa_{aa}}{\rho_{cc}/\rho_{aa}}}\right)^{-1}, \qquad (2)$$

where $\kappa_{ii}$ and $\rho_{ii}$ are the thermal conductivity and the electrical resistivity along the $i$-axis directions, respectively. In an ideal goniopolar material with isotropic electrical conductivity and thermal conductivity, this angle is 45°. The transverse voltage disappears when the heat-current direction coincides with the crystallographic axis.

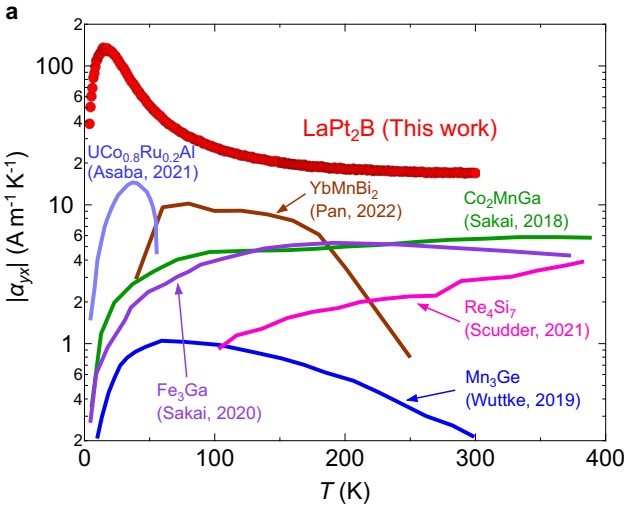

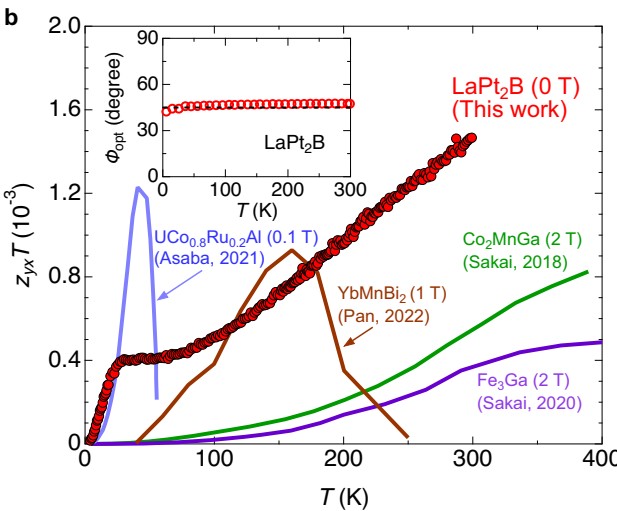

**Fig. 5 | Giant transverse Peltier conductivity and transverse thermoelectric performance. a** Temperature dependence of the transverse Peltier conductivity $|\alpha_{yx}|$ of LaPt$_2$B compared with those of the ANE-based systems (UCo$_{0.8}$Ru$_{0.2}$Al[16], YbMnBi$_2$[17], Co$_2$MnGa[15], Fe$_3$Ga[8], Mn$_3$Ge[45]) and a goniopolar conductor (Re$_4$Si$_7$[21]). **b** Comparison of transverse thermoelectric figure of merit ($z_{yx}T$) of several

transverse thermoelectric systems. The $z_{yx}T$ of LaPt$_2$B was evaluated using Eq. (3) in a zero magnetic field. The measured magnetic field used for the evaluation of $z_{yx}T$ is shown for other transverse thermoelectric materials[8,15–17,46]. The inset shows the temperature dependence of $\phi_{opt}$ of LaPt$_2$B estimated using Eq. (2).

Therefore, achieving practical and stable transverse thermoelectric conversion requires the optimal angle to be close to 45°. It should be noted that this optimal angle in highly anisotropic compounds such as layered systems becomes far from 45°; for example, when $\rho_{cc}/\rho_{aa} \sim 10^3$, the optimal angle is estimated to $\phi_{opt} \approx 10°$, provided $\kappa_{cc}/\kappa_{aa} = 1$. In the case of LaPt$_2$B, $\phi_{opt}$ is approximately 45° (Fig. 5b) due to the isotropic $\rho$ and $\kappa$ in the temperature range below room temperature (Fig. 2d and Fig. S10), indicating that LaPt$_2$B is an ideal goniopolar material. Generally, $\phi_{opt}$, reflecting the temperature variation of anisotropy in transport properties, exhibits complex temperature dependence. The temperature-invariant $\phi_{opt}$ in LaPt$_2$B demonstrates the ability to achieve the maximum transverse thermoelectric conversion efficiency under a fixed geometry.

We finally evaluated the dimensionless figure of merit ($z_{yx}T$) for the transverse thermoelectricity[5] of LaPt$_2$B as

$$z_{yx}T = \frac{(S_{cc} - S_{aa})^2 T}{(\sqrt{\rho_{cc}\kappa_{cc}} + \sqrt{\rho_{aa}\kappa_{aa}})^2} = \frac{4S_{yx}^2 T}{(\sqrt{\rho_{cc}\kappa_{cc}} + \sqrt{\rho_{aa}\kappa_{aa}})^2}. \quad (3)$$

Figure 5b shows the comparison of the transverse $z_{yx}T$ of LaPt$_2$B and several transverse thermoelectric systems. The transverse $z_{yx}T$ of LaPt$_2$B exhibits a large value as high as $1.5 \times 10^{-3}$ at room temperature, which is approximately three times larger than the room-temperature $z_{yx}T$ in the full-Heusler topological magnet[15]. The mixed-dimensional conductor LaPt$_2$B exhibits outstanding transverse thermoelectric performance in zero magnetic fields, while the external magnetic field or remanent magnetization is necessary for ANE-based systems.

In summary, we demonstrated the large transverse thermopower (10 μV K$^{-1}$ at $T$ = 300 K) and Peltier conductivity (130 A K$^{-1}$ m$^{-1}$ at $T$ = 15 K) in the mixed-dimensional conductor LaPt$_2$B. Materials with axis-dependent thermopower polarity, capable of achieving significant transverse thermoelectricity even at zero magnetic fields, offer substantial practical advantages compared to ANE-based systems. In addition, the utilization of the mixed-dimensionality of stable Fermi surfaces in metals for transverse thermoelectricity provides practical benefits by enabling solid transverse thermoelectric conversion in a wide temperature range from room temperature to extremely low temperatures. The substantiated mixed-

dimensional conductor in this study paves the way for exploring efficient transverse thermoelectric conversion materials.

## Methods
### Single-crystal growth
LaPt$_2$B polycrystal wes synthesized via arc melting using a tetra arc furnace under an argon atmosphere. The starting materials were La (99.9%), Pt (99.95%), and B (99.9%) with a stoichiometric amounts. The ingot was flipped and remelted five times to ensure homogeneity. Single crystals of LaPt$_2$B were grown from the polycrystalline ingot using the Czochralski method in a tetra-arc furnace under an argon atmosphere. The single crystal was pulled from the melting ingot at a speed of 12 mm h$^{-1}$.

The crystal structure was confirmed using powder and single-crystal XRD. The measured single crystals were oriented using a Laue camera (Photonic Science, X-ray Laue Back-Scattered Camera, Fig. S1).

### Transport measurement
Electrical resistivity was measured using a conventional dc four-probe method with a Keithley 2182A nanovoltmeter. The excitation current of $I$ = 1 mA was provided by a Keithley 6221 current source. Thermopower for both crystallographic axes was measured using a steady-state technique with manganin-constantan differential thermocouples[37,38] in a closed-cycle refrigerator. The thermoelectric voltage was measured with a Keithley 2182A nanovoltmeter. The temperature gradient with a typical value of 0.5 K mm$^{-1}$ was applied using a resistive heater. The thermoelectric voltage from the wire leads was subtracted. Transverse thermopower measurements were performed using four manganin-constantan thermocouples. Voltage drop and temperature difference were measured simultaneously for both longitudinal ($x$) and transverse ($y$) directions. The typical sample size used for the measurements is approximately 1.5 mm × 1.5 mm × 0.5 mm. The results for transverse thermopower measurements of a reference sample are shown in Supplementary information (Fig. S7).

### Thermal image
The thermal image was taken using an infrared thermography camera (FSV-1200, Apiste) at room temperature. A black body coating was applied to correct the emissivity of the sample. This measurement was

performed using the same sample used for the transverse thermo-electric power measurements.

## First-principles calculations

First-principles calculations based on density functional theory (DFT) were performed using Quantum Espresso[39,40]. We used the projector-augmented-wave (PAW) pseudopotentials with the Perdew-Burke-Ernzerhof (PBE) exchange-correlation functional. The cut-off energies for plane waves and charge densities were set to 80 and 640 Ry, respectively, and the $k$-point mesh was set to $20 \times 20 \times 20$ uniform grid to ensure the convergence. The density of states (DOS) was obtained using the optimized tetrahedron method[41]. We performed fully rela-tivistic calculations with spin-orbit coupling (SOC) in addition to the scalar relativistic calculations. The calculated Fermi surfaces are drawn using FermiSurfer program[42].

The transport coefficients were calculated based on the linearized Boltzmann equations under relaxation time approximation by using BoltzTraP2 code[43], in which a smoothed Fourier interpolation was implemented to achieve dense $k$-point grid[44]. The transport coeffi-cients were calculated only for the scalar relativistic case in BoltzTraP2 code. Using the energy of the $n$-th band at $\mathbf{k}$ point $E_{n,\mathbf{k}}$, the transport distribution function tensor $\Sigma_{ij}(\varepsilon)$ is given as

$$\Sigma_{ij}(\varepsilon) = \sum_n \Sigma_{ij}^n(\varepsilon) = \sum_n \sum_{\mathbf{k}} v_i v_j \tau \delta(\varepsilon - E_{n,\mathbf{k}}), \quad (4)$$

where $\Sigma_{ij}^n(\varepsilon)$ is the partial transport distribution function tensor of $n$-th band, $v_i$ is the $i$-th component of the band velocity $\mathbf{v} = \frac{1}{\hbar}\nabla_{\mathbf{k}}E_{n,\mathbf{k}}$, $\tau$ is the relaxation time, and $\delta$ is the delta function.

Then, the transport coefficients are evaluated as a function of the chemical potential $\mu$ for the temperature $T$. The partial electrical conductivity tensor of $n$-th band $\sigma_{ij}^n(\mu)$ is

$$\sigma_{ij}^n(\mu) = e^2 \int_{-\infty}^{\infty} d\varepsilon \left(-\frac{\partial f_0}{\partial \varepsilon}\right) \Sigma_{ij}^n, \quad (5)$$

where $e$ is the elementary charge and $f_0$ is the Fermi-Dirac distribution function. The total electrical conductivity tensor $\sigma_{ij}(\mu)$ is

$$\sigma_{ij}(\mu) = \sum_n \sigma_{ij}^n(\mu) = e^2 \int_{-\infty}^{\infty} d\varepsilon \left(-\frac{\partial f_0}{\partial \varepsilon}\right) \Sigma_{ij}. \quad (6)$$

Similarly, the partial Peltier conductivity tensor of $n$-th band $\alpha_{ij}^n(\mu) = [\sigma S]_{ij}^n(\mu)$ is

$$\alpha_{ij}^n(\mu) = -\frac{e}{T} \int_{-\infty}^{\infty} d\varepsilon \left(-\frac{\partial f_0}{\partial \varepsilon}\right)(\varepsilon - \mu)\Sigma_{ij}^n, \quad (7)$$

where $S_{ij}$ is the thermopower tensor. The total Peltier conductivity is given as

$$\alpha_{ij}(\mu) = \sum_n \alpha_{ij}^n(\mu) = -\frac{e}{T} \int_{-\infty}^{\infty} d\varepsilon \left(-\frac{\partial f_0}{\partial \varepsilon}\right)(\varepsilon - \mu)\Sigma_{ij}. \quad (8)$$

Chemical potential dependence of the diagonal components $\sigma_{ii}(\mu)$ and $\alpha_{ii}(\mu)$ are shown in the Supplementary information (Fig. S4). The calculation of the transport coefficients relies on a calculated band structure at $T = 0$ K. We here ignore the temperature dependence of the relative band energies due to the electron-phonon interaction. The thermopower $S_{ii}$ for the $i$-axis direction is obtained as $S_{ii} = \alpha_{ii}/\sigma_{ii}$.

The inverse mass tensor of the $n$-th band is obtained using the calculated energy of the $n$-th band at $\mathbf{k}$ point $E_{n,\mathbf{k}}$ as

$$\left[\frac{1}{m^*}\right]_{ij} = \frac{1}{\hbar^2}\frac{\partial^2 E_{n,\mathbf{k}}}{\partial k_i \partial k_j}, \quad (9)$$

indicating that the positive (negative) value of the inverse mass tensor expresses electron-like (hole-like) band curvature.

## Data availability

The data that support the findings of this study are available from the corresponding author upon reasonable request.

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

## Acknowledgements

This work was supported by JSPS KAKENHI (22K20360, 22H01166), GIMRT Program (202205-IRKAC-0049, 202212-IRKMA-0005), and Research Foundation for the Electrotechnology of Chubu (REFEC, No. R-04102).

## Author contributions

Y.J.S. conceived and planned the project. H.M., S.O. and Y.J.S. carried out the transport measurements and analyzed the data. H.M., Y.J.S. and R.O. performed the first-principles calculations. Y.J.S. and D.A. synthe-sized the high-quality single-crystal samples. Y.J.S. prepared the manuscript with input from all the authors.

## Competing interests

The authors declare no competing interests.
