## [Peer Review File · Nature Communications]

Large transverse thermoelectric effect induced by the mixed-dimensionality of Fermi surfacesREVIEWER COMMENTS

Reviewer #1 (Remarks to the Author):

attached

Reviewer #2 (Remarks to the Author):

This paper reports on systematic investigation of transverse thermoelectric conversion properties of a single-crystalline LaPt₂B. The experimental results show that LaPt₂B exhibits clear goniopolarity driven by multidimensional electronic structures. The observed transverse Peltier conductivity is more than an order of magnitude larger than typical values for the anomalous Nernst effect, although its figure of merit is small. The goniopolarity is one of the promising principles for transverse thermoelectric conversion, but material candidates are extremely limited. Since this study provides a new candidate for goniopolar materials, it is important for both fundamental and applied studies on thermoelectrics. I can thus recommend the publication of this paper if the authors address the following issues.

1) As described on the caption of Fig. 2, the first-principles calculation of the Seebeck coefficient was performed with fixing the relaxation time at 10^{-14} s. How did the authors determine the value of the relaxation time and confirm its validity? Is the temperature-independent relaxation time enough to discuss the thermoelectric properties of LaPt₂B?

2) The large transverse Peltier conductivity is attributed to the very small electrical resistivity of LaPt₂B, not to its transverse thermopower. This point should be emphasized clearly. The authors should discuss the microscopic origin of the very small electrical resistivity of LaPt₂B in more detail.

3) As shown in Fig. 3a, the transverse thermopower increases with increasing temperature. The authors should discuss the possibility of using LaPt₂B at high temperatures. What is the maximum temperature at which this material can be used in terms of stability and temperature dependence of transport coefficients?

4) Since the Seebeck coefficient of LaPt₂B is small, the measured thermopower is affected by the Seebeck effect of electrical leads. How did the authors remove the contribution of the Seebeck coefficient of the leads to obtain the absolute Seebeck coefficient of LaPt₂B?

5) Estimation of the figure of merit is important to discuss the potential of LaPt₂B as a thermoelectric material. Thus, Fig. S12 should be moved to the main text. It is unfair that the figure of merit for Re₄Si₇ is removed in Fig. S12c,d, which is much larger than that for LaPt₂B.

6) English should be rechecked. For example, the phrase "To analyze more quantitatively" has no object.

7) Figure S9 is not cited in the text of Supplementary Information.

Reviewer #3 (Remarks to the Author):

(See attached referee report word document.)

This work reports the gonipolar thermoelectric properties of LaPt2B. Seebeck coefficient with opposite signs are observed along different crystallographic directions, which is attributed to the highly anisotropic electron and hole Fermi pockets as demonstrated by theoretical calculations. Along a 45 degree direction rotated from crystallographic axis, a large transverse thermopower, and a giant transverse thermoelectric conductivity, as well as a competitive transverse zT are achieved under zero magnetic field. The paper is well organized and written. The major issue of this paper is that the large values of thermoelectric conductivity and zT are not calculated directly from measuring the transport properties from x, y directions. Specific comments are listed below.

- (1) The authors have shown the measurement setup for S_{yx} , and S_{yx} is measured by applying the temperature gradient along the x direction. However, the authors calculate the thermoelectric conductivity using $\alpha_{yx} = (\sigma_{aa}S_{aa} - \sigma_{cc}S_{cc})/2$ (for 45 degree condition), instead of calculating the α_{yx} from $\alpha_{yx} = \sigma_{yx}S_{xx} + \sigma_{yy}S_{yx}$. The authors can simply measure σ_{yx} and σ_{yy} and resolve the α_{yx} directly. Though theoretically $\alpha_{yx} = (\sigma_{aa}S_{aa} - \sigma_{cc}S_{cc})/2$ looks correct, as an experimental work, it's always better that the key values are obtained directly from experiments. I wonder if there is a specific reason why the authors did not calculate the α_{yx} from $\alpha_{yx} = \sigma_{yx}S_{xx} + \sigma_{yy}S_{yx}$.
- (2) This is the same case for the zT calculation, why not simply measure the resistivity along y direction and thermal conductivity along the x direction, and then calculate zT from $zT_{yx} = (S_{yx}^2/(\rho_{yy} \kappa_{xx}))T$?
- (3) I suggest the authors to measure the Hall resistivity along a- and c-axis. A gonipolar material shall also present a Hall coefficient with opposite sign along different crystallographic axis, which can make the present work more convincing.
- (4) Can the authors comment the physical reason for the large thermoelectric conductivity and Peltier angle? For example, the large thermoelectric conductivity in ANE is usually attributed to the topological Berry phase. What can be the physics behind such a large thermoelectric conductivity that orders higher than the Berry phase effect?
- (5) Figure 2 shows the Fermi surfaces calculated in the absence of spin-orbit coupling. Why not present the Fermi surfaces with SOC taken into account, since LaPt2B should have a large SOC?
- (6) From my point of view, the title of the present work is a bit of confusing, especially the word "multi-dimensional". I understand the coexistence of the 1D monoaxial chiral spin structure and 2D layer crystal structure in LaPt2B, however, the word "multi-dimensional" makes readers confused at first glance.

Referee report on Manako, *et al.* “Orthogonal thermoelectricity in a multi-dimensional goniopolar conductor”, Nature Communications manuscript #652463

This manuscript reports the observation of $p \times n$ -type transverse thermoelectricity in a new material, LaPt₂B, whereby p -type Seebeck coefficient is observed in an orthogonal direction to n -type Seebeck coefficient. The manuscript argues that attention to the effective dimensionality of the respective electron and hole surfaces is novel, and that when the dimensions of the Fermi surfaces are 1D and 2D, respectively, the transverse thermoelectric performance is optimized. Band structure calculations are presented to show 5 bands that cross the Fermi surface, of which two are proposed to dominate a 1D conduction in the valence band along the c -axis and 2D conduction in the electron band along the a - b plane. The Peltier conductivity is proposed as an experimental and theoretical metric of dimensionality relevant to thermoelectric anisotropy, and experimental results are proposed to be qualitatively consistent with theoretical prediction.

The results will have impact in the field, largely because single-crystal transverse thermoelectrics (TTE) are still a developing field, and this paper introduces a new TTE material with large transverse power-factor. However, there are a series of conclusions drawn by the authors which, as presented, are insufficiently supported. Below we point these out, and in some cases suggest possible ways to address these insufficiencies.

Insufficiently supported or insufficiently referenced claims requiring edits:

- 1) Claim of “Multi-dimensionality”: The consideration of band dimensionality in optimizing TTE performance that the authors refer to as “multi-dimensional” is, in fact, not novel. The paper that first introduced the importance of considering dimensionality of the respective bands was the original manuscript that introduced TTE behavior in single-crystal materials, namely Zhou, PRL (2013) {Ref. [2] in this manuscript} which explains TTE transport as a 2D-like hole band conducting in parallel with a 3D-like valence band, and explicitly names bulk crystals like Re₄Si₇ and PdCoO₂ as bulk crystals whose anisotropic band structures can also explain such behavior. Subsequent theoretical work by the same authors emphasized the importance of identifying dimensionality in conduction and valence bands by solving the Boltzmann transport equation for conductivity and Seebeck tensors in 1D, 2D, and 3D bands {Grayson, InTechOpen (2018) <https://www.intechopen.com/chapters/62183>, not referenced in this manuscript}.
 - Recommendation: Rather than propose that “multi-dimensionality” is a new concept, give proper historical background as supporting evidence that the band dimensionality is an important design strategy.
 - Rather than the somewhat confusing and ambiguous name “multi-dimensionality” (which could be misinterpreted as simply more than one dimension), an alternate name “mixed-dimensionality” seems more appropriate, given that mixed conduction refers to electron and hole conduction, so mixed dimensionality implies that the dimensionality of the electron and hole bands might be different.

- The band structure of WSi₂ {Koster, Chem. Mater. (2023), Ref. 10 in this manuscript} is a *far* more convincing example of mixed dimensionality (Fig. 2 reproduced here), with a 1D-like flat electron band, panel (c), and a 2D-like open-cylindrical hole band, panel (e).

This material is an appropriate example of an ideal mixed-dimensional band structure as proposed in Fig. 1(e) of the present manuscript, whose characteristics are then to be sought out in the band structure of other materials, such as that introduced in this manuscript, LaPt₂B.

- However, LaPt₂B should not be identified as having “*firm evidence of its multi-dimensional nature*” (line 144), nor can “*the coexistence of hole and electron Fermi surfaces with different dimensionality*” be considered “*crucially important to realize the goniopolarity in this system*” (line 120). *Any* anisotropic crystal with mixed electron-hole conduction and anisotropically oriented Fermi surfaces of *any* non-spherical shape will exhibit $p \times n$ behavior. Mixed dimensionality is not by any

means a “*crucial*” requirement. In fact, the Fermi surfaces in this very manuscript as shown in Fig. 2(f) have significant deviations from the simplified 1D + 2D example of Fig. 1. In particular, the alpha band in Fig. 2(f) is a closed 3D hexagonal Fermi surface with a hole in the middle, not an open planar 1D Fermi surface as shown in Fig. 1(e); and the gamma band in Fig. 2(f) has a puckered open cylindrical Fermi surface with a hexagonal toothed Fermi surface around its waist, not a flat open cylindrical 2D Fermi surface as shown in Fig. 1(e). These obvious deviations need to be explicitly mentioned, not completely ignored.

- One possible solution would be for the authors to argue that the identification of features that *somewhat* resemble the open 1D and open 2D Fermi surfaces might lead to TTE behavior. If so, these Fermi surfaces should be labelled as *quasi-1D* and *quasi-2D* (not “1D” and “2D”), and the authors could make the argument that such resemblance to the ideal case is still a compelling reason why such materials should be explored as possible TTE candidates.
- But most importantly, the arguments given by the authors that these Fermi surfaces *are* quasi-1D and quasi-2D is incorrect at face value. The primary evidence cited by the authors that the Fermi surfaces are 1D or 2D is the anisotropic Peltier conductivity, but this results from an integral over the entire Fermi surface. The authors did not provide any breakdown of the different pieces of the Fermi surface to show how MUCH of the *p*-type Peltier conductivity of the alpha band arises from the features of interest, *i.e.*, how much of the 1D-like flat top and bottom surfaces of the hexagon account for the majority of the *p*-type behavior and how much of the 2D-like warped cylindrical part of the electron Fermi surface accounts for the majority of the *n*-type behavior. Without such a breakdown, the main conclusion of the paper that “mixed dimensionality” is a useful rule of thumb for identifying TTE’s is completely unsupported. However, *with* such an analysis, the authors could argue the validity of their hypothesis.

2) Measurement of transverse Seebeck voltage: In the measurement geometry of the transverse Seebeck voltage shown in Fig. 3(d) and (e) the authors make the statement in line 150, “the temperature gradient $-\nabla T$ is homogeneously applied to 45° from the *a*-axis to the *c*-axis.” Although the authors then interpret the temperature gradient as uniform, it is, in fact, not expected to be uniform given that the edges of the sample have significant overlap with the heat source and heat sink terminals. This will cause the actual thermal gradient in the relevant region of the sample center to be LARGER than the average value, and thus the estimated Seebeck may be greater than the actual Seebeck. [Note that with LONGITUDINAL thermoelectrics, the exact shape of the thermal distribution is irrelevant since one integrates the electric field over the same path as the thermal gradient. However, with TRANSVERSE thermoelectrics, the transverse electric field integrates over a lateral path, whereas the thermal gradient integrates over a longitudinal path.]

- Please provide a cross-sectional view of the temperature map of Fig. 3(e), plotted as temperature versus position. From this more proper analysis of the temperature gradient in the middle of the sample one can recalculate S_{yx} . It may be, for example,

that the circa 30% difference between the measured S_{yx} and theoretical one in Fig. 3(b) might be due to this inhomogeneous temperature gradient.

- It must be specified WHICH transverse S_{yx} value (the red curve in Fig. 3(b) or the green curve in Fig. 3 (b)) is used to calculate the experimental transverse Peltier conductivity α_{yx} and transverse power factor. The direct experimentally measured S_{yx} measured in the 45° rotated sample in Fig. 3(d,e) is less accurate than the S_{yx} deduced from the rotation of the longitudinal S_{xx} and S_{yy} measurements for the reasons specified above. The latter S_{yx} is deduced from rotation of the diagonal Seebeck tensor and is more accurate and should be use for calculating all transverse power factors and Peltier conductivities.
- Fig. 3(a) is misleading in this regard, given that S_{yx} can only be determined from an exact knowledge of the local thermal gradient in the middle of the sample at all temperatures. Instead, this referee suggests that the vertical axis in Fig. 3(a) should be plotted as a Seebeck VOLTAGE V_{yx} in microvolts versus temperature, and Fig. 3(b) can be used to show consistency of this voltage estimate (red) with the more exactly deduced transverse voltage (green). The GREEN curve deduced from rotation is the more correct estimate of the transverse Seebeck coefficient and should be used in all power-factor calculations.

- 3) Comparison of Peltier angle among different materials: In Fig. 4(b), great effort is made to compare different materials with respect to the angle between the charge and heat currents, which the authors define as the “Peltier angle” and which the authors seem to imply is a materials property. However, note that this angle is NOT a materials property has already been previously defined under the name θ_{JQ} in the paper of Qing, *et al.* MRS Advances, 4, 497 (2018) {not referenced in this manuscript}. It is not a materials property since the angle θ_{JQ} can be 90° in *any transverse thermoelectric material* with proper choice of current angle in the material ϕ , as explained in Qing *et al.* In fact, there are up to two different solutions for the charge current angle in any TTE θ_{JQ} to achieve perfect $\theta_{JQ} = 90^\circ$ angle with respect to the heat current, one for the current angle $0^\circ < \phi \leq 45^\circ$ and one for $45^\circ \leq \phi < 90^\circ$. Thus, the entire Fig. 4(b) and discussion surrounding it emphasizing the $\theta_{JQ} \sim 90^\circ$ transverse behavior seem to be misguided.
- This referee recommends that this discussion be stricken or at least de-emphasized. The authors only ever consider a current angle of $\phi = 45^\circ$ with respect to their crystal axes in their samples, and therefore seem to arrive at the conclusion that their Peltier angle θ_{JQ} being close to 90° is somehow special, when it is, in fact, a simple consequence of the magnitude of the *p*-Seebeck coefficient along the *c*-axis being approximately equal to the magnitude of the *n*-Seebeck coefficient along the *a*-*b* plane (see Eqs. (8) and (10) in Qing, *et al.*). For all of the other TTE’s shown in Fig. 4(b) there exist current angles of ϕ relative to the crystal axes for which they would achieve $\theta_{JQ} = 90^\circ$. The entire discussion is therefore misleading and unremarkable since the Peltier angle, as such, is NOT a materials property, it depends on the relative direction of the current with respect to the crystal axes.

- 4) Discussion in Paragraph #2, page 1: The language and terms used in the second paragraph on p. 1 are confusing, ambiguous, and, at times, incompletely referenced. For example, the fact that the transverse thermoelectric effect can originate from anisotropic mobilities in electron and hole bands was first proposed in Zhou et al. {Ref. [2] in the manuscript}, which is not cited here. Also, this paragraph would be clearer if it explicitly differentiated the two-band transverse thermoelectric effect (which goes under two names: " $p \times n$ TTE's" and "two-band goniopolar," according to different authors) from the single-band TTE's (called "goniopolar bands"). This paragraph should be rewritten for clarity. A lot of the grammar and logic is difficult to follow.

Recommended edits:

- 5) Figure 1 is slightly confusing since 4 diagrams are included which describe "Longitudinal," "Transverse," "Nernst," and "Goniopolar," whereas the latter two are both examples of the second "Transverse" case. The recommendation is to eliminate the "Transverse" diagram entirely and put the prefix "Transverse" in front of the labels for the two transverse examples. (This referee recommends that the TTEs be labelled with the original name given to this class of materials upon discovery, namely, "Transverse $p \times n$," rather than the rebranded name "goniopolar".) Note, also, that the point width of the lines that indicate which vector arrows correspond with which variables are too narrow to see.
- 6) The simulated Seebeck coefficients in Fig. 2(e) rely on a band structure calculation, however, the band structure is presumably calculated at $T = 0$, and typical bands may undergo a relative shift of order 50-100 meV when temperature is increased from 0 K to 300 K due to electron-phonon renormalization which empirically goes under the name of the "Varshni effect" {see, for example, Ning, et al. Phys. Chem. Chem. Phys. 25, 26006 (2023).} To this end, the authors should explicitly state their assumption that there is no temperature dependence to the relative band energies up to room temperature. In a semimetal such as LaPt₂B, the shifting band energies would cause changes in both the electron and hole densities, affecting the density of states around the Fermi energy and therefore the Seebeck coefficient.
- 7) In the Supplemental Info Fig. S12(c), the log-scale is misleading. ZT on the vertical axis needs to be in scientific notation without factoring out 10^{-3} .
- 8) For the Fig. 4 Anomalous Nernst Effect, if this comparison is to be made, then the range of saturation magnetic fields for these ANE materials should also be listed in the caption and/or the body text, to make clear that ANE requires an external B-field, whereas TTE does not.
- 9) In Fig. S8 a) the authors need to specify at what angle ϕ these calculations are performed. Presumably 45° , but this needs to be explicitly stated.

10) Typo in Fig. S10 caption “aixs”.

Citation corrections:

Note 1: Nature Communications might not allow citations in the abstract, in which case the citations in the abstract should be renumbered and referenced in the order they appear within the body of the manuscript. Note that some citations (such as [1] Goldsmid) might not appear again elsewhere in the manuscript as-written, so care should be taken to re-insert these as appropriate in the body of the paper.

Note 2: Historically, Ref. 2, Zhou, *et al.* PRL (2013), is the first in the literature to propose that transverse thermoelectric behavior can be observed in materials with p-type Seebeck orthogonal to n-type Seebeck. In that respect, the mixed-electron/hole conduction model, such as that proposed in this paper, can make mention of the original name applied to this thermoelectric functionality, namely “ $p \times n$ -type transverse thermoelectrics” before introducing the more commonly applied name “goniopolar” to which the same materials were later rebranded. The majority of recent TTE literature has neglected to cite Zhou, *et al.* as the proper genesis of the field, and it would be commendable for the current manuscript to address this systematic error in the literature.

Note 3: The following $p \times n$ -type transverse thermoelectric is missing from the citation list, so the authors might not be aware of this work: Cohn, *et al.*, PRL 112, 186602 (2014) studying the quasi-one-dimensional metal $\text{Li}_{0.9}\text{Mo}_6\text{O}_{17}$.

Reply to the Reviewer #1 -- #652463 Manako *et al.*

Thank you very much for your report and helpful comments. We have carefully considered your comments and revised the manuscript as follows.

- (1) The authors have shown the measurement setup for S_{yx} , and S_{yx} is measured by applying the temperature gradient along the x direction. However, the authors calculate the thermoelectric conductivity using $\alpha_{yx} = (\sigma_{aa}S_{aa} - \sigma_{cc}S_{cc})/2$ (for 45 degree condition), instead of calculating the α_{yx} from $\alpha_{yx} = \sigma_{yx}S_{xx} + \sigma_{yy}S_{yx}$. The authors can simply measure σ_{yx} and σ_{yy} and resolve the α_{yx} directly. Though theoretically $\alpha_{yx} = (\sigma_{aa}S_{aa} - \sigma_{cc}S_{cc})/2$ looks correct, as an experimental work, it's always better that the key values are obtained directly from experiments. I wonder if there is a specific reason why the authors did not calculate the α_{yx} from $\alpha_{yx} = \sigma_{yx}S_{xx} + \sigma_{yy}S_{yx}$.

Response: We have measured σ_{xx} and σ_{yy} . On the other hand, when rotating the current direction from the crystallographic axes, measuring the off-diagonal components (σ_{xy} and σ_{yx}) is very challenging due to the small absolute value and the issues related to terminal misalignment. Note that the term $\sigma_{yx}S_{xx}$ in the transverse Peltier conductivity becomes very small due to both the small σ_{yx} and the small S_{xx} in an isotropic goniopolar conductor. Therefore, in the goniopolar conductor, $\sigma_{yy}S_{yx}$ becomes dominant in the evaluation of α_{yx} . As shown in the figure below, we plotted the transverse Peltier conductivity estimated from several methods (Eq.(7) and Eq (9) in supplementary information, $\alpha_{yx} = 0.5 * (\sigma_{aa} + \sigma_{cc}) * S_{yx}$, and $\alpha_{yx} = \sigma_{yy} * S_{yx}$). The transverse Peltier conductivity estimated using these methods is of comparable magnitude, indicating the validity of the transverse Peltier conductivity estimation in this study. We have added the figure below in supplementary information (Page 12-13, Fig. S12, in SI).

(2) This is the same case for the zT calculation, why not simply measure the resistivity along y direction and thermal conductivity along the x direction, and then calculate zT from $zT_{yx} = (S_{yx}^2/(\rho_{yy} \kappa_{xx}))T$?

Response: We measured ρ_{yy} and κ_{xx} (x and y are directions rotated 45 degrees from the a-axis and c-axis) and evaluated $z_{yx}T$ using the expression of $z_{yx}T = (S_{yx}/(\rho_{yy}\kappa_{xx}))T$ (Orange markers in the figure below). The $z_{yx}T$ estimated using the expression of $z_{yx}T = (S_{yx}/(\rho_{yy}\kappa_{xx}))T$ exhibits slightly different behavior compared to the $z_{yx}T$ estimated using other expressions (the second and third terms in Eq. (3) in the manuscript), as shown in the figure below. The difference in the estimation of $z_{yx}T$ may be attributed to measurement errors in κ_{xx} . Accurate measurement of κ_{xx} is difficult due to issues related to the sample's geometry. For the reader's information, the comparison of $z_{yx}T$ evaluated using these expressions has been added to the supplementary information (Page 16, Fig. S15, in SI).

(3) I suggest the authors to measure the Hall resistivity along *a*- and *c*-axis. A goniopolar material shall also present a Hall coefficient with opposite sign along different cryptographic axis, which can make the present work more convincing.

Response: We agree with the reviewer's suggestion. We measured Hall effect for the electrical current along *a*- and *c*-axis, as shown in the figures below. Panel *a* shows the field dependence of the Hall resistivity for *H* || *a*, *J* || *c* and *H* || *c*, *J* || *a*. The estimated Hall coefficient is negative for both *J* || *a* and *J* || *c*. The sign of the Hall coefficient for *J* || *a*, *E* || *b* (*b* is the in-plane direction perpendicular to the *a*-axis and *c*-axis), *H* || *c* is consistent with the results of Seebeck coefficient for in-plane direction. On the other hand, the sign of R_H for *J* || *c*, *E* || *b*, *H* || *a* is reversed compared to the sign of the Seebeck coefficient for the *c*-axis. This sign reversal is attributed to the influence of the multi-carrier effect, as the R_H is evaluated in the *cb*-plane (*c* and *b* direction are the out-of-plane and in-plane directions, respectively). The experimentally obtained sign and magnitude of R_H are in good agreement with those calculated using the BoltzTraP2 code. We have added the experimental and calculated results of the Hall effect to the supplementary information (Page 10, Fig. S11 in SI).

(4) Can the authors comment the physical reason for the large thermoelectric conductivity and Peltier angle? For example, the large thermoelectric conductivity in ANE is usually attributed to the topological Berry phase. What can be the physics behind such a large thermoelectric conductivity that orders higher than the Berry phase effect?

Response: As shown in Fig. 3b, the large significant transverse thermoelectric effect in the goniopolar conductor LaPt₂B can be explained based on the rotation of the thermopower tensor. The large transverse Peltier conductivity α_{yx} is attributed to the large transverse thermopower in the goniopolar conductor and the high electrical conductivity

of a good metal. In particular, the significant increase of low-temperature α_{yx} is due to the large electrical conductivity of high-quality single crystal of LaPt₂B and the phonon drag enhancement of the Seebeck coefficient. The fact that LaPt₂B exhibits a Peltier angle close to 90 degrees indicates that LaPt₂B is an ideal goniopolar conductor, namely the thermopower with the same magnitude but opposite sign, along with isotropic electrical and thermal conductivity. In addition, the almost 45-degree optimal angle, which maximizes transverse thermoelectric performance, also indicates the ideal goniopolarity of LaPt₂B.

(5) Figure 2 shows the Fermi surfaces calculated in the absence of spin-orbit coupling. Why not present the Fermi surfaces with SOC taken into account, since LaPt₂B should have a large SOC?

Response: As pointed out by the reviewer, spin-orbit coupling plays a crucial role in the electronic structure of LaPt₂B with a chiral structure. We performed first-principles calculations including SOC, and we show the obtained Fermi surfaces in Fig. S3 (supplementary information). While the first-principles calculation including SOC leads to small changes in in the Fermi surface topology, the anisotropy of Fermi surfaces remains crucial for the goniopolar conduction in LaPt₂B, as evidenced by the results of the transport property calculations (Fig. S5, in SI).

(6) From my point of view, the title of the present work is a bit of confusing, especially the word “multi-dimensional”. I understand the coexistence of the 1D monoaxial chiral spin structure and 2D layer crystal structure in LaPt₂B, however, the word “multi- dimensional” makes readers confused at first glance.

Response: We appreciate the comment. The term “multi-dimensional” might be confusing for readers at first glance. In the revised manuscript, we have replaced “multi-dimensionality” with “mixed-dimensionality”. We have also changed the title of the paper to “Large transverse thermoelectric effect in a mixed-dimensional goniopolar conductor”.

Reply to the Reviewer #2 -- #652463 Manako *et al.*

Thank you very much for your report and helpful comments. We have carefully considered your comments and revised the manuscript as follows.

- (1) As described on the caption of Fig. 2, the first-principles calculation of the Seebeck coefficient was performed with fixing the relaxation time at 10^{-14} s. How did the authors determine the value of the relaxation time and confirm its validity? Is the temperature-independent relaxation time enough to discuss the thermoelectric properties of LaPt₂B?

Response: Since the calculation of the relaxation time is very difficult task owing to the many-body effect, we here adopt a constant relaxation time approximation. The value of $\tau = 10^{-14}$ sec. is a default value in the BoltzWann software. So far, several studies have used the default relaxation time for transport property calculations [for example, K. Berland *et al.*, Appl. Phys. Lett. 119, 081902 (2021)].

The Seebeck coefficient is calculated from the expression $S_{ii} = \alpha_{ii}/\sigma_{ii}$, where the σ is the electrical conductivity and α is the Peltier conductivity. Based on the Boltzmann transport equations, σ_{ii} and α_{ii} can be calculated as Eq. (3) and Eq. (5) (the methods section in the manuscript), respectively. Both σ_{ii} and α_{ii} include the transport distribution function defined in Eq. (1) in the methods section in the manuscript. The transport distribution function includes the relaxation time, and the relaxation time is canceled out when calculating the Seebeck coefficient. Indeed, the thermopower is approximately expressed without the relaxation time as $S \sim (k_B/e)(k_B T/E_F)$ in metals and $S \sim (k_B/e)(\Delta/k_B T)$ (Δ is the band gap) in insulators [K. Behnia, *Fundamentals of Thermoelectricity*, Oxford University Press (2015)].

Also note that, although the relaxation time is temperature-dependent, we compare the band-resolved Peltier conductivity in Fig. 3b at the fixed temperature to discuss the carrier polarity and the dimensionality of each band. Such band-resolved Peltier conductivity under constant relaxation time approximation has also been discussed in a multiband system to examine the Fermi surface character [J. S. You *et al.*, Phys. Rev. B **103**, 045102 (2021)].

- (2) The large transverse Peltier conductivity is attributed to the very small electrical resistivity of LaPt₂B, not to its transverse thermopower. This point should be emphasized clearly. The authors should discuss the microscopic origin of the very small electrical resistivity of LaPt₂B in more detail.

Response: We agree with the reviewer that the large transverse Peltier conductivity is attributed to the small electrical resistivity of LaPt₂B. LaPt₂B is a good metal with large Fermi surface and large carrier number leads to very small electrical resistivity of LaPt₂B. We have revised the sentence regarding the large transverse Peltier conductivity. (Page 5, LL. 216-219) However, we would say the transverse thermopower of LaPt₂B (10 $\mu\text{V}/\text{K}$ @300 K, 0 T) is also sufficiently large, compared to the ANE systems: Co₂MnGa (6 $\mu\text{V}/\text{K}$ @300 K, 2 T, [A. Sakai *et al.*, Nat. Phys. (2018)]), Fe₃Ga (5.5 $\mu\text{V}/\text{K}$ @300 K, 2 T, [A. Sakai *et al.*, Nature (2020)]) and YbMnBi₂ (6 $\mu\text{V}/\text{K}$ @160 K, 1 T, [Y. Pan *et al.*, Nat. Mater. (2022)]). The low-temperature α_{yx} is largely enhanced due to the large low-temperature electrical conductivity in high-quality single crystal as well as the phonon-drag enhancement of the Seebeck coefficient.

- (3) As shown in Fig. 3a, the transverse thermopower increases with increasing temperature. The authors should discuss the possibility of using LaPt₂B at high temperatures. What is the maximum temperature at which this material can be used in terms of stability and temperature dependence of transport coefficients?

Response: As pointed out by the reviewer, LaPt₂B is a metal, and it is expected that the transverse thermopower will increase linearly in the high-temperature range. The transport properties calculations (Fig. S5 in SI) also support the increase of the transverse thermopower at high temperatures. LaPt₂B has a melting point exceeding 1000 °C, but oxidization may occur at high temperatures in the air. We plan to measure the high-temperature Seebeck coefficient of LaPt₂B under the vacuum condition for the future study.

- (4) Since the Seebeck coefficient of LaPt₂B is small, the measured thermopower is affected by the Seebeck effect of electrical leads. How did the authors remove the contribution of the Seebeck coefficient of the leads to obtain the absolute Seebeck coefficient of LaPt₂B?

Response: We measured thermopower using manganin-constantan thermocouples. Figure below shows the schematic diagram of the steady-state method circuit used in this study. A temperature difference is applied between two points on the sample and the thermocouple, and this temperature difference between the two points is denoted as ΔT . Here the potential difference ΔV_{Man} measured by manganin wires can be written as follows:

$$\begin{aligned} \Delta V_{\text{Man}} &= \int_{T_{\text{Room}}}^{T'} (-S_{\text{lead}}) dT + \int_{T'}^T (-S_{\text{Man}}) dT + \int_T^{T+\Delta T} (-S_{\text{smp}}) dT + \int_{T+\Delta T}^{T'} (-S_{\text{Man}}) dT + \int_{T'}^{T_{\text{Room}}} (-S_{\text{lead}}) dT + V_0 \\ &= - \int_T^{T+\Delta T} (S_{\text{smp}} - S_{\text{Man}}) dT + V_0 \end{aligned}$$

, where S_{smp} , S_{lead} , S_{Man} , S_{Con} , and V_0 are the Seebeck coefficients of the sample, lead wire, manganin wire, constantan wire, and an offset voltage, respectively. Therefore, when the temperature dependence of $S_{\text{smp}} - S_{\text{Man}}$ is negligible within the range of the temperature difference, the ΔV_{Man} can be expressed as $\Delta V_{\text{Man}} = -(S_{\text{smp}} - S_{\text{Man}})\Delta T + V_0$. We apply two different temperature differences to remove the offset voltage V_0 . Finally, the Seebeck coefficient of the sample was calculated by correcting the known Seebeck coefficient of manganin. The Seebeck coefficients in several reference samples such as the constantan and high-Tc cuprates were checked in this measurement system.

- (5) Estimation of the figure of merit is important to discuss the potential of LaPt₂B as a thermoelectric material. Thus, Fig. S12 should be moved to the main text. It is unfair that the figure of merit for Re₄Si₇ is removed in Fig. S12c,d, which is much larger than that for LaPt₂B.

Response: We thank the reviewer for this comment. We have added a figure showing the comparison of dimensionless figure of merit ($z_{yx}T$) in the main text. In the paper on the transverse thermoelectric properties of Re₄Si₇ [M. R. Scudder *et al.*, Energy Env. Sci. (2021)], the estimated $z_{yx}T$ based on the transverse thermopower measurements has been reported above 400 K. In addition, thermal conductivity of Re₄Si₇ has been reported only above 300 K, preventing the estimation of zT below 300 K. Therefore, due to the difficulty in reasonable estimation for zT of Re₄Si₇, we have not plotted that zT data for Re₄Si₇ on the figure.

- (6) English should be rechecked. For example, the phrase "To analyze more quantitatively" has no object.

Response: We appreciate the comment. We have made English revisions throughout the entire manuscript.

- (7) Figure S9 is not cited in the text of Supplementary Information.

Response: We thank the reviewer for this comment. We have revised the paragraph related to Fig. S9 and the sample dependence of LaPt₂B (Page 8, in SI).

Reply to the Reviewer #3 -- #652463 Manako *et al.*

We thank the reviewer for time reviewing our manuscript and helpful comments. We have carefully considered your comments and revised the manuscript as follows.

(1) Claim of “Multi-dimensionality”: The consideration of band dimensionality in optimizing TTE performance that the authors refer to as “multi-dimensional” is, in fact, not novel. The paper that first introduced the importance of considering dimensionality of the respective bands was the original manuscript that introduced TTE behavior in single- crystal materials, namely Zhou, PRL (2013) {Ref. [2] in this manuscript} which explains TTE transport as a 2D-like hole band conducting in parallel with a 3D-like valence band, and explicitly names bulk crystals like Re₄Si₇ and PdCoO₂ as bulk crystals whose anisotropic band structures can also explain such behavior. Subsequent theoretical work by the same authors emphasized the importance of identifying dimensionality in conduction and valence bands by solving the Boltzmann transport equation for conductivity and Seebeck tensors in 1D, 2D, and 3D bands {Grayson, InTechOpen (2018) <https://www.intechopen.com/chapters/62183>, not referenced in this manuscript}.

- Recommendation: Rather than propose that “multi-dimensionality” is a new concept, give proper historical background as supporting evidence that the band dimensionality is an important design strategy.

- Rather than the somewhat confusing and ambiguous name “multi-dimensionality” (which could be misinterpreted as simply more than one dimension), an alternate name “mixed-dimensionality” seems more appropriate, given that mixed conduction refers to electron and hole conduction, so mixed dimensionality implies that the dimensionality of the electron and hole bands might be different.

Response: We appreciate the comment. We have made overall revisions to the introduction, emphasizing the importance of considering dimensionality of the respective bands, as discussed in previous studies (Page 1, second paragraph).

We agree with the comment that the term “multi-dimensional” might be confusing for readers. In the revised manuscript, we have replaced “multi-dimensionality” with “mixed-dimensionality”, as suggested by the referee. We have also changed the title of the paper to “Large transverse thermoelectric effect in a mixed-dimensional goniopolar conductor”.

- The band structure of WSi₂ {Koster, Chem. Mater. (2023), Ref. 10 in this manuscript} is a *far* more convincing example of mixed dimensionality (Fig. 2 reproduced here), with a 1D-like flat electron band, panel (c), and a 2D-like open-cylindrical hole band, panel (e). This material is an appropriate example of an ideal mixed-dimensional band structure as proposed in Fig. 1(e) of the present manuscript, whose characteristics are then to be sought out in the band structure of other materials, such as that introduced in this manuscript, LaPt₂B.

Response: We have revised the third paragraph to emphasize that the mixed-dimensionality of the Fermi surface in metals is important as a design principle for efficient transverse thermoelectric materials (Page 1, third paragraph). In addition, we have added further details regarding the important previous study of WSi₂ (Page 1, third paragraph).

- However, LaPt₂B should not be identified as having “firm evidence of its multi-dimensional nature” (line 144), nor can “the coexistence of hole and electron Fermi surfaces with different dimensionality” be considered “crucially important to realize the goniopolarity in this system” (line 120). *Any* anisotropic crystal with mixed electron-hole conduction and anisotropically oriented Fermi surfaces of *any* non-spherical shape will exhibit p×n behavior. Mixed dimensionality is not by any means a “crucial” requirement. In fact, the Fermi surfaces in this very manuscript as shown in Fig. 2(f) have significant deviations from the simplified 1D + 2D example of Fig. 1. In particular, the alpha band in Fig. 2(f) is a closed 3D hexagonal Fermi surface with a hole in the middle, not an open planar 1D Fermi surface as shown in Fig. 1(e); and the gamma band in Fig. 2(f) has a puckered open cylindrical Fermi surface with a hexagonal toothed Fermi surface around its waist, not a flat open cylindrical 2D Fermi surface as shown in Fig. 1(e). These obvious deviations need to be explicitly mentioned, not completely ignored.

- One possible solution would be for the authors to argue that the identification of features that *somewhat* resemble the open 1D and open 2D Fermi surfaces might lead to TTE behavior. If so, these Fermi surfaces should be labelled as *quasi-1D* and *quasi-2D* (not “1D” and “2D”), and the authors could make the argument that such resemblance to the ideal case is still a compelling reason why such materials should be explored as possible TTE candidates.

- But most importantly, the arguments given by the authors that these Fermi surfaces *are* quasi-1D and quasi-2D is incorrect at face value. The primary evidence cited by the authors that the Fermi surfaces are 1D or 2D is the anisotropic Peltier conductivity,

but this results from an integral over the entire Fermi surface. The authors did not provide any breakdown of the different pieces of the Fermi surface to show how MUCH of the p-type Peltier conductivity of the alpha band arises from the features of interest, i.e., how much of the 1D-like flat top and bottom surfaces of the hexagon account for the majority of the p-type behavior and how much of the 2D-like warped cylindrical part of the electron Fermi surface accounts for the majority of the n-type behavior. Without such a breakdown, the main conclusion of the paper that “mixed dimensionality” is a useful rule of thumb for identifying TTE’s is completely unsupported. However, *with* such an analysis, the authors could argue the validity of their hypothesis.

Response: We thank the reviewer for this comment. We have added the discussion on the inverse effective mass tensor onto the Fermi surfaces to examine the relation between the anisotropy of the band-resolved partial Peltier conductivity (Page 4, third paragraph in Mixed-dimensional Fermi surfaces subsection of RESULTS section and Fig. 3 c-f). The figure below is Fig. 3 in the revised manuscript. Panels c-f show the in-plane and out-of-plane components of the inverse mass tensor projected on the mixed-dimensional Fermi surfaces α and γ . The quasi-1D FS α exhibits a significant contribution from hole for the out-of-plane direction compared to the in-plane direction. In contrast, the cylindrical part near the A point of the γ FS exhibits a strong contribution from electron for the in-plane direction, indicating the quasi-2D nature of the γ Fermi surface. Here we discuss the inverse effective mass tensor instead of the effective mass tensor because the strength of the inverse effective mass is significant for the transport properties. The analysis of the inverse mass tensor provides firm evidence for the correlation between the mixed-dimensional topology of the Fermi surfaces and the anisotropy of the Peltier conductivity. This analysis supports the conclusion that the goniopolar conductivity in LaPt₂B is attributed to the mixed-dimensional Fermi surface.

- (2) Measurement of transverse Seebeck voltage: In the measurement geometry of the transverse Seebeck voltage shown in Fig. 3(d) and (e) the authors make the statement in line 150, “the temperature gradient $-\nabla T$ is homogeneously applied to 45° from the a -axis to the c -axis.” Although the authors then interpret the temperature gradient as uniform, it is, in fact, not expected to be uniform given that the edges of the sample have significant overlap with the heat source and heat sink terminals. This will cause the actual thermal gradient in the relevant region of the sample center to be LARGER than the average value, and thus the estimated Seebeck may be greater than the actual Seebeck. [Note that with LONGITUDINAL thermoelectrics, the exact shape of the thermal distribution is irrelevant since one integrates the electric field over the same path as the thermal gradient. However, with TRANSVERSE thermoelectrics, the transverse electric field integrates over a lateral path, whereas the thermal gradient integrates over a longitudinal path.]
- Please provide a cross-sectional view of the temperature map of Fig. 3(e), plotted as temperature versus position. From this more proper analysis of the temperature gradient in the middle of the sample one can recalculate S_{yx} . It may be, for example, that the circa 30% difference between the measured S_{yx} and theoretical one in Fig. 3(b) might be due to this inhomogeneous temperature gradient.

Response: We thank the reviewer for this comment. We agree with the reviewer that the uniformity of the temperature gradient is important. As shown in the figure below, we investigated the position dependence of temperature measured using an infrared thermography camera. The yellow markers indicate the positions of the thermocouples. An almost linear temperature dependence is observed along the longitudinal (x) direction, while the temperature variation is very small along the transverse (y) direction. This indicates that the heat flow is applied one-dimensionally along the longitudinal direction, and the temperature gradient is uniform in the present measurement system. Based on the thermograph data, there is a maximum error of approximately 10 % in the local temperature gradient, and it is considered that the transverse thermopower measurements in this study were performed with sufficient accuracy. On the other hand, the thermal imaging measurement was performed only above room temperature, because the black-body radiation rapidly decreases at low temperatures. Therefore, we added the statement to assume a homogeneous temperature gradient in the revised manuscript (Page 4, right column). We have added the graph of position-dependence of temperature measured using a thermography camera in Fig. 4f (Page 5) in the main text.

- It must be specified WHICH transverse S_{yx} value (the red curve in Fig. 3(b) or the green curve in Fig. 3 (b)) is used to calculate the experimental transverse Peltier conductivity α_{yx} and transverse power factor. The direct experimentally measured S_{yx} measured in the 45° rotated sample in Fig. 3(d,e) is less accurate than the S_{yx} deduced from the rotation of the longitudinal S_{xx} and S_{yy} measurements for the reasons specified above. The latter S_{yx} is deduced from rotation of the diagonal Seebeck tensor and is more accurate and should be use for calculating all transverse power factors and Peltier conductivities.

- Fig. 3(a) is misleading in this regard, given that S_{yx} can only be determined from an exact knowledge of the local thermal gradient in the middle of the sample at all temperatures. Instead, this referee suggests that the vertical axis in Fig. 3(a) should be plotted as a Seebeck VOLTAGE V_{yx} in microvolts versus temperature, and Fig. 3(b) can be used to show consistency of this voltage estimate (red) with the more exactly deduced transverse voltage (green). The GREEN curve deduced from rotation is the more correct estimate of the transverse Seebeck coefficient and should be used in all power-factor calculations.

Response: As pointed out by the reviewer, we have replaced the main panel of Fig. 4a (Fig. 3a in the old manuscript) with the temperature dependence of the transverse Seebeck voltage $-\Delta V_y$. As mentioned later, since we evaluate the transverse thermoelectric performance using directly measured S_{yx} , we have retained the graph of the temperature dependence of S_{yx} as an inset. We note that the voltage generated by the transverse thermoelectric effect is proportional to the longitudinal temperature difference, and it is not a materials property. Green curves in Fig. 4b were calculated using the longitudinal temperature difference ΔT_x . We evaluated the transverse Peltier conductivity and transverse thermoelectric performance based on the experimentally determined S_{yx} . The S_{yx} was calculated using the directly measured ΔV_y and ΔT_x . We have mentioned that we calculate the Peltier conductivity and zT based on the directly measured S_{yx} in the main text. We acknowledge the reviewer's concern that experimentally measured S_{yx} is less accurate than calculated one due to the inhomogeneity of the temperature gradient. However, the aforementioned analysis of the temperature gradient using the thermograph supports that the measurement accuracy is enough in this study. Note that measurement errors due to the uniformity of heat flow are also present in conventional thermoelectric measurements. As pointed out by reviewer #1, we also believe that, as an experimental work, it makes sense to discuss the transverse thermoelectric performance based on directly measured data. We have added the sentences to explain the details of evaluation for the Peltier conductivity (Page 5, LL. 213-215 in the revised manuscript).

(3) Comparison of Peltier angle among different materials: In Fig. 4(b), great effort is made to compare different materials with respect to the angle between the charge and heat currents, which the authors define as the “Peltier angle” and which the authors seem to imply is a materials property. However, note that this angle is NOT a materials property has already been previously defined under the name θ_{JQ} in the paper of Qing, et al. MRS Advances, 4, 497 (2018) {not referenced in this manuscript}. It is not a materials property since the angle θ_{JQ} can be 90° in any transverse thermoelectric material with proper choice of current angle in the material ϕ , as explained in Qing et al. In fact, there are up to two different solutions for the charge current angle in any TTE θ_{JQ} to achieve perfect $\theta_{JQ} = 90^\circ$ angle with respect to the heat current, one for the current angle $0^\circ < \phi \leq 45^\circ$ and one for $45^\circ \leq \phi < 90^\circ$. Thus, the entire Fig. 4(b) and discussion surrounding it emphasizing the $\theta_{JQ} \sim 90^\circ$ transverse behavior seem to be misguided.

- This referee recommends that this discussion be stricken or at least de-emphasized. The authors only ever consider a current angle of $\phi = 45^\circ$ with respect to their crystal axes in their samples, and therefore seem to arrive at the conclusion that their Peltier angle θ_{JQ} being close to 90° is somehow special, when it is, in fact, a simple consequence of the magnitude of the p-Seebeck coefficient along the c-axis being approximately equal to the magnitude of the n-Seebeck coefficient along the a-b plane (see Eqs. (8) and (10) in Qing, et al.). For all of the other TTE's shown in Fig. 4(b) there exist current angles of ϕ relative to the crystal axes for which they would achieve $\theta_{JQ} = 90^\circ$. The entire discussion is therefore misleading and unremarkable since the Peltier angle, as such, is NOT a materials property, it depends on the relative direction of the current with respect to the crystal axes.

Response: I appreciate the comments from the reviewer. As follows the reviewer's comment, we have removed the plot of the transverse Peltier conductivity against the θ_{JQ} from the main text in the revised manuscript. Instead, we have added the plot of the dimensionless figure of merit as Fig. 4b.

The reviewer's comment about the presence of a heat flow angle where θ_{JQ} becomes 90 degrees in goniopolar conductors is valid. However, in transverse thermoelectric materials, the actual heat flow direction and optimal angle exhibit complex temperature dependence, arising from the temperature variation in anisotropy of thermal conductivity and the Seebeck coefficient. Therefore, evaluating θ_{JQ} with a fixed angle of heat flow is a useful parameter for discussing the performance of transverse thermoelectric materials. LaPt₂B can be considered an ideal goniopolar conductor because the value of θ_{JQ}

remains close to 90 degrees in a wide temperature range when the angle of heat flow is fixed at 45 degrees from the crystallographic axes. The fact that the optimal angle of LaPt₂B is approximately 45 degrees across the wide temperature range also supports that LaPt₂B is an ideal goniopolar conductor. We have added the plot of the Peltier conductivity versus Peltier angle in supplementary information (Page 14-15, Fig. S14, in SI).

- (4) Discussion in Paragraph #2, page 1: The language and terms used in the second paragraph on p. 1 are confusing, ambiguous, and, at times, incompletely referenced. For example, the fact that the transverse thermoelectric effect can originate from anisotropic mobilities in electron and hole bands was first proposed in Zhou et al. {Ref. [2] in the manuscript}, which is not cited here. Also, this paragraph would be clearer if it explicitly differentiated the two-band transverse thermoelectric effect (which goes under two names: “p × n TTE’s and “two-band goniopolar,” according to different authors) from the single-band TTE’s (called “goniopolar bands”). This paragraph should be rewritten for clarity. A lot of the grammar and logic is difficult to follow.

Response: As suggested by the reviewer, we have revised the second paragraph on page 1. Considering the historical context, we have explicitly mentioned the term “p × n-type conductor” in the revised manuscript. We have also mentioned the term “single-band goniopolar” in the introduction section. In the recent review by Uchida and Heremans, the term “goniopolar” is defined as anisotropic electron/hole conduction due to intrinsic crystalline structure, and we have avoided using the term “two-band goniopolar” in the revised manuscript to avoid confusion.

- (5) Figure 1 is slightly confusing since 4 diagrams are included which describe “Longitudinal,” “Transverse,” “Nernst,” and “Goniopolar”, whereas the latter two are both examples of the second “Transverse” case. The recommendation is to eliminate the “Transverse” diagram entirely and put the prefix “Transverse” in front of the labels for the two transverse examples. (This referee recommends that the TTEs be labelled with the original name given to this class of materials upon discovery, namely, “Transverse p × n,” rather than the rebranded name “goniopolar”.) Note, also, that the point width of the lines that indicate which vector arrows correspond with which variables are too narrow to see.

Response: As suggested by the reviewer, we have removed the second “Transverse” diagram from Fig. 1 to clarify the comparison between longitudinal and transverse thermoelectricity. Additionally, We have added the label “ $p \times n$ -type” to Fig.1c (Page 2, Figure 1). We also adjusted the width of the lines in Fig.1c.

- (6) The simulated Seebeck coefficients in Fig. 2(e) rely on a band structure calculation, however, the band structure is presumably calculated at $T = 0$, and typical bands may undergo a relative shift of order 50-100 meV when temperature is increased from 0 K to 300 K due to electron-phonon renormalization which empirically goes under the name of the “Varshnii effect” {see, for example, Ning, et al. Phys. Chem. Chem. Phys. 25, 26006 (2023).} To this end, the authors should explicitly state their assumption that there is no temperature dependence to the relative band energies up to room temperature. In a semimetal such as LaPt₂B, the shifting band energies would cause changes in both the electron and hole densities, affecting the density of states around the Fermi energy and therefore the Seebeck coefficient.

Response: Thank you for your comment. The band structure is indeed calculated at $T = 0$ K. In the revised manuscript, we noted our assumption that there is no temperature dependence of the relative band energies due to the electron-phonon interaction.

- (7) In the Supplemental Info Fig. S12(c), the log-scale is misleading. ZT on the vertical axis needs to be in scientific notation without factoring out 10^{-3} .

Response: We agree with the reviewer’s comment. In the revised manuscript, we have added the plot of zT in the main text, and we have removed the plot of zT on a semi-logarithmic scale in the revised supplementary information.

- (8) For the Fig. 4 Anomalous Nernst Effect, if this comparison is to be made, then the range of saturation magnetic fields for these ANE materials should also be listed in the caption and/or the body text, to make clear that ANE requires an external B-field, whereas TTE does not.

Response: Thank you for the reviewer’s comment. We have included the magnitude of the external magnetic field in Fig. 4. In addition, we have explicitly mentioned that an external magnetic field is required for the ANE-based system, while it is not necessary

for the pn -type thermoelectric materials (Page 4-5, LL. 204-205 , Page 6, Fig. 5b).

(9) In Fig. S8 a) the authors need to specify at what angle ϕ these calculations are performed. Presumably 45° , but this needs to be explicitly stated.

Response: As mentioned by the reviewer, the solid lines in Fig. S8a are calculated S_{yx} and S_{xx} for $\phi = 45^\circ$. We have revised the caption of Fig. S8 (Page 8, in SI).

(10) Typo in Fig. S10 caption “aixs”.

Response: We appreciate the comment. We have corrected the typo (Page 9, in SI).

Citation corrections:

Note 1: Nature Communications might not allow citations in the abstract, in which case the citations in the abstract should be renumbered and referenced in the order they appear within the body of the manuscript. Note that some citations (such as [1] Goldsmid) might not appear again elsewhere in the manuscript as-written, so care should be taken to re-insert these as appropriate in the body of the paper.

Note 2: Historically, Ref. 2, Zhou, et al. PRL (2013), is the first in the literature to propose that transverse thermoelectric behavior can be observed in materials with p-type Seebeck orthogonal to n-type Seebeck. In that respect, the mixed-electron/hole conduction model, such as that proposed in this paper, can make mention of the original name applied to this thermoelectric functionality, namely “p × n -type transverse thermoelectrics” before introducing the more commonly applied name “goniopolar” to which the same materials were later rebranded. The majority of recent TTE literature has neglected to cite Zhou, et al. as the proper genesis of the field, and it would be commendable for the current manuscript to address this systematic error in the literature.

Note 3: The following p × n -type transverse thermoelectric is missing from the citation list, so the authors might not be aware of this work: Cohn, et. al., PRL 112, 186602 (2014) studying the quasi-one-dimensional metal $\text{Li}_{0.9}\text{Mo}_6\text{O}_{17}$.

Response: We have revised the references based on the reviewer’s comment.

Summary of Changes -- #652463 Manako *et al.*

Revised manuscript

Title: We have changed the title of the paper to “Large transverse thermoelectric effect in a mixed-dimensional goniopolar conductor”.

Main text: We have replaced “multi-dimensionality” with “mixed-dimensionality” in the abstract and revised main text.

Page 1, Abstract: We have revised the abstract based on revisions to the main text. We have explicitly mentioned the term “ $p \times n$ -type” in the abstract, introduction, and summary. We have removed the reference numbers in the abstract.

Page 2, LL. 61-63: We have revised the entire of second paragraph to explain the importance of considering dimensionality of the respective bands, as discussed in previous studies.

Page 2, LL. 27-60: We have revised the third paragraph to emphasize that the mixed-dimensionality of the Fermi surface in metals is important as a design principle for efficient transverse thermoelectric material.

Page 2, Figure 1: We have removed the second “Transverse” diagram from Fig. 1 to clarify the comparison between longitudinal and transverse thermoelectricity. Additionally, We have added the label “ $p \times n$ -type” to Fig.1c. We also adjusted the width of the lines in Fig.1c.

Page 2-3, LL. 80-88: We have added further details regarding the important previous study of WSi_2 .

Page 3, Figure 2: We have removed panels 2f and 2g from Fig.2.

Page 4, Figure 3: We have added a new figure as Fig. 3. (Figures 3 and 4 in the old paragraph are Figs. 4 and 5 in the revised manuscript.) Figure 3 includes the panels 2f and 2g in the old paragraph and a new panel which shows the strength of the inverse mass tensor.

Page 4, LL. 161-175: We have added the discussion on the inverse effective mass tensor onto the Fermi surfaces to examine the relation between the anisotropy of the band-resolved

partial Peltier conductivity.

Page 4, LL. 192-200, Page 5, Figure 4: We have replaced the main panel of Fig. 4a (Fig. 3a in the old manuscript) with the temperature dependence of the transverse Seebeck voltage $-\Delta V_y$. We have shown the graph of the temperature dependence of S_{yx} as an inset. We have added the graph of position-dependence of temperature measured using a thermography camera as Fig. 4f.

Page 4-5, LL. 204-205: We have explicitly mentioned that an external magnetic field is not necessary for the $p \times n$ -type thermoelectric materials.

Page 5, LL. 213-215: We have revised the sentences to explain the details of evaluation for the Peltier conductivity.

Page 5, LL. 216-219: We have revised the sentence regarding the large transverse Peltier conductivity.

Page 5-6, LL. 223-262, Figure 5: We have removed the plot of Peltier conductivity versus Peltier angle from Fig. 5 (Fig. 4 in the old manuscript). We have added the temperature dependence of $z_{yx}T$ and optimal angle as Fig. 5c.

Page 7, References: We have added references as follows:

[21] Cohn, J. L. *et al.*, Phys. Rev. Lett. **112**, 186602 (2014).

[29] Grayson, M. *et al.*, Bringing Thermoelectricity into Reality pp. 81-100 (2018).

Page 8 LL. 457-460: We have added the sentences to explain transport coefficients calculations as follows: "The calculation of the transport coefficients relies on a calculated band structure at $T = 0$ K. We here ignore the temperature dependence of the relative band energies due to the electron-phonon interaction."

Page 8 LL. 462-465: We have added the sentences describing the calculation of the inverse mass tensor.

Supplementary information

Page 8: We have revised the paragraph related to Fig. S9.

Page 10, Section H, Fig. S11: We have added the experimental and calculated results of the Hall effect of LaPt₂B.

Page 12-13, Fig. S12: We have added sentences and a figure to discuss the validity of the evaluation of the Peltier conductivity.

Page 14-15, Section K, Fig. S14: We have added the plot of the Peltier conductivity versus Peltier angle. We have added a paragraph to discuss the comparison of LaPt₂B and other transverse thermoelectric systems.

Page 15-16, Fig. S15: We have added sentences and a figure to discuss the validity of the evaluation of the dimensionless figure of merit.

Page 7, References: We have added references.

REVIEWER COMMENTS

Reviewer #1 (Remarks to the Author):

The authors have addressed most of the concerns. However, the authors still have some work to do regarding calling LaPt2B a goniopolar material. A goniopolar material shall present a Hall coefficient (also a Seebeck coefficient) with opposite sign along different cryptographic axis. Since the Hall resistivity along a- and c-axis are both negative, the different sign of the Seebeck coefficient can be simply originate from anisotropic properties of both electrons and holes, i.e., $S_{total} = (S_e \sigma_e + S_h \sigma_h) / (\sigma_e + \sigma_h)$, wherein S and σ are Seebeck coefficient and electrical conductivity, and e, h represents electrons and holes, respectively. In this case, maybe it is better to call LaPt2B a mixed-dimensional (which guarantee the strong anisotropy that results in opposite sign of the Seebeck coefficient) conductor instead of a goniopolar conductor. The authors measured the Hall coefficient along a and c axis at different temperatures, why not keep the temperature the same? Could the authors calculate the Seebeck coefficient along both a and c axis from $S_{x_total} = (S_{x_e} \sigma_{x_e} + S_{x_h} \sigma_{x_h}) / (\sigma_{x_e} + \sigma_{x_h})$? (x = a and c), to check if the different signs of the Seebeck coefficient along a and c simply come from the anisotropic transport behavior?

Reviewer #2 (Remarks to the Author):

The authors have addressed my comments appropriately, and the manuscript is improved. This paper is ready for publication if the other reviewers are also satisfied with the revisions.

Reply to the Reviewer #1 -- #652463 Manako *et al.*

Thank you very much for your report and helpful comments. We have carefully considered your comments and revised the manuscript as follows.

The authors have addressed most of the concerns. However, the authors still have some work to do regarding calling LaPt₂B a goniopolar material. A goniopolar material shall present a Hall coefficient (also a Seebeck coefficient) with opposite sign along different cryptographic axis. Since the Hall resistivity along a- and c-axis are both negative, the different sign of the Seebeck coefficient can be simply originate from anisotropic properties of both electrons and holes, i.e., $S_{total} = (S_e \sigma_e + S_h \sigma_h) / (\sigma_e + \sigma_h)$, wherein S and σ are Seebeck coefficient and electrical conductivity, and e, h represents electrons and holes, respectively. In this case, maybe it is better to call LaPt₂B a mixed-dimensional (which guarantee the strong anisotropy that results in opposite sign of the Seebeck coefficient) conductor instead of a goniopolar conductor.

Response: We appreciate the reviewer's suggestion regarding the term "goniopolar," which refers to a single-band picture for the axis-dependent conduction polarity. We revisited this point and modified it to use the term "mixed-dimensional conductor" instead of "goniopolar conductor" in the revised manuscript. We have included the sentences to explain the concept of "mixed-dimensional conductor" (Page 3, LL. 119-129).

The authors measured the Hall coefficient along a and c axis at different temperatures, why not keep the temperature the same? Could the authors calculate the Seebeck coefficient along both a and c axis from $S_{x,total} = (S_{x,e} \sigma_{x,e} + S_{x,h} \sigma_{x,h}) / (\sigma_{x,e} + \sigma_{x,h})$? (x = a and c), to check if the different signs of the Seebeck coefficient along a and c simply come from the anisotropic transport behavior?

Response: We measured the Hall coefficient for both crystallographic axes at 10 K (Supplementary information, Page 10).

The total thermopower $S_{x,total}$ along the x-axis (x = a and c) is generally given as $S_{x,total} = (\sum_n \alpha_{x,n}) / (\sum_n \sigma_{x,n})$, where $\alpha_{x,n} = S_{x,n} \sigma_{x,n}$ is the band-resolved partial Peltier conductivity and n is the band index. We have calculated the partial Peltier conductivity and the total thermopower (Figs. 3b and 2e) and found that the alpha sheet has a large positive partial Peltier conductivity along the c axis, indicating a quasi-1D conduction of holes on the alpha sheet. In contrast, the gamma sheet has a negative partial Peltier conductivity along a axis, indicating a quasi-2D conduction of electrons. Thus, the mixed-dimensionality is crucial for the axis-dependent thermopower polarity. On the other hand, since the conduction polarities coincide along the a and c axes for each sheet (Fig. 3b), we avoid the use of the term goniopolar for the present material as suggested by the Referee. The measured Hall coefficient quantitatively agrees with that calculated based on the energy band structure (Fig. S11 in Supplementary information). This result also indicates the mixed-dimensionality is important for the axis-dependent thermopower polarity of LaPt₂B, rather than single-band anisotropy.

Reply to the Reviewer #2 -- #652463 Manako *et al.*

The authors have addressed my comments appropriately, and the manuscript is improved. This paper is ready for publication if the other reviewers are also satisfied with the revisions.

We thank the reviewer for recommending publication and the constructive feedback throughout the reviewing process.

Summary of Changes -- #652463 Manako *et al.*

Revised manuscript

Title: We have changed the title of the paper to “Large transverse thermoelectric effect induced by the mixed-dimensionality of Fermi surfaces”.

Main text: We have replaced “axis-dependent conduction polarity” with “axis-dependent thermopower polarity” for LaPt₂B in the abstract and revised main text. In addition the term “goniopolar” was replaced with “mix-dimensional conductor” .

Page 3, LL. 119-129: We have included the sentences to explain the concept of “mixed-dimensional conductor”.

Supplementary information

Page 10, Section H, Fig. S11(a): We have updated the figure using experimental data of the Hall effect of LaPt₂B measured at 10 K.

Page 10, Section H, LL. 1-2: We have added the description of measurement details.